# ATMOSSCI-BENCH: Evaluating the Recent Advances of Large Language Models for Atmospheric Science

**Chenyue Li**
CSE, HKUST
Clear Water Bay, Hong Kong SAR
clieh@connect.ust.hk

**Wen Deng**
CE, HKUST
Clear Water Bay, Hong Kong SAR
wdengan@connect.ust.hk

**Mengqian Lu**
CE, HKUST
Clear Water Bay, Hong Kong SAR
cemlu@ust.hk

**Binhang Yuan**\*
CSE, HKUST
Clear Water Bay, Hong Kong SAR
biyuan@ust.hk

## Abstract

The rapid advancements in large language models (LLMs), particularly in their reasoning capabilities, hold transformative potential for addressing complex challenges and boosting scientific discovery in atmospheric science. However, leveraging LLMs effectively in this domain requires a robust and comprehensive evaluation benchmark. Toward this end, we present ATMOSSCI-BENCH, a novel benchmark designed to systematically assess LLM performance across five core categories of atmospheric science problems: hydrology, atmospheric dynamics, atmospheric physics, geophysics, and physical oceanography. ATMOSSCI-BENCH features a dual-format design comprising both multiple-choice questions (MCQs) and open-ended questions (OEQs), enabling scalable automated evaluation alongside deeper analysis of conceptual understanding. We employ a template-based MCQ generation framework to create diverse, graduate-level problems with symbolic perturbation, while OEQs are used to probe open-ended reasoning. We conduct a comprehensive evaluation of representative LLMs, categorized into four groups: instruction-tuned models, advanced reasoning models, math-augmented models, and domain-specific climate models. Our analysis provides some interesting insights into the reasoning and problem-solving capabilities of LLMs in atmospheric science. We believe ATMOSSCI-BENCH can serve as a critical step toward advancing LLM applications in climate services by offering a standard and rigorous evaluation framework. The source code of ATMOSSCI-BENCH is available at [https://github.com/Relaxed-System-Lab/AtmosSci-Bench].

## 1 Introduction

Large language models (LLMs) [1], especially in their reasoning capabilities, have recently achieved remarkable progress, offering transformative potential for addressing complex challenges in atmospheric science [2, 3, 4, 5]. More recently, increasingly powerful LLMs have accelerated progress in AI4S (AI for Science), enabling a paradigm shift in scientific discovery. With their growing capabilities, LLMs show the potential to act as "AI Scientists," partially assisting—or even autonomously conducting—hypothesis generation, experimental design, execution, analysis, and refinement [6, 7, 8, 9, 10, 11]. To advance AI for Atmospheric Science and enable the development of reliable and effective LLM-based applications for climate-related tasks, it is crucial to recognize

---

\*Corresponding author.

39th Conference on Neural Information Processing Systems (NeurIPS 2025) Track on Datasets and Benchmarks.

that LLMs themselves serve as a foundational core. Assessing whether current LLMs are capable of reasoning about problems in this domain is therefore a prerequisite, which calls for *a robust and comprehensive evaluation framework. Such a benchmark is essential to systematically assess the performance of LLMs across a diverse array of atmospheric science problems*, ensuring their utility, accuracy, and robustness in this critical domain.

Atmospheric science presents unique and complex challenges, ranging from micro-scale processes like cloud dynamics to global-scale climate systems. To ensure that LLMs can effectively contribute to solving these real-world problems, it is essential to establish a benchmark that evaluates their performance, especially their reasoning and interpretative abilities. Such a well-designed benchmark will not only foster innovation but also provide a standardized framework for assessing the utility, accuracy, and robustness of LLMs in this field.

Atmospheric science problems include essential differences from the classic mathematical and physical problems commonly found in existing LLM benchmarks [12, 13]. This field is inherently interdisciplinary, requiring the integration of theoretical analytical modeling skills with real-world phenomena and knowledge. Concretely, atmospheric science involves analyzing and synthesizing heterogeneous data types, such as spatial coordinates, temperatures, wind patterns, and empirical estimates, which are often presented in varied formats and units. Furthermore, solving these problems necessitates the selection of appropriate physical models and mathematical methods to ensure accuracy, adding layers of complexity beyond traditional benchmarks. As such, constructing a benchmark tailored to atmospheric science is a necessary complement to existing evaluations, enabling a more comprehensive assessment of LLMs' reasoning capabilities.

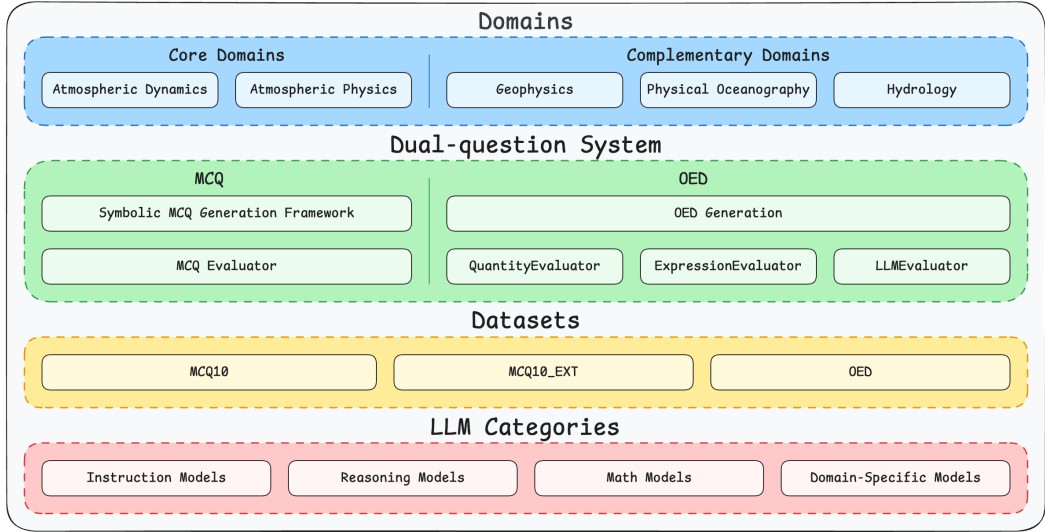

Figure 1: Overview of ATMOSSCI-BENCH

To address this need, we introduce ATMOSSCI-BENCH, a benchmark designed to comprehensively evaluate recent advances of LLMs in atmospheric science and related domains. An overview of ATMOSSCI-BENCH is shown in Figure 1. The construction process—including the dual-question system, datasets, and domain coverage—is described in Section 3, while the evaluation and categories of LLMs assessed are detailed in Section 4. Concretely, we summarize our key contributions:

**Contribution 1.** We construct ATMOSSCI-BENCH, a comprehensive benchmark comprising Multiple-Choice Questions (MCQs) and Open-Ended Questions (OEQs) to assess LLM performance across five scientific domains relevant to atmospheric science: (i) *atmospheric dynamics*, (ii) *atmospheric physics*, (iii) *geophysics*, (iv) *hydrology*, and (v) *physical oceanography*. MCQs enable scalable automated assessment via symbolic templates, while OEQs reveal deeper LLM reasoning ability in open-ended settings. The question set is curated from graduate-level materials and systematically expanded using a symbolic generation framework to ensure relevance and diversity.

**Contribution 2.** We conduct a comprehensive evaluation that includes a wide range of representative LLMs, which can be concretely categorized into four classes: (i) *instruction model*s that have been

fine-tuned for instruction following; (ii) *reasoning models* that have been aligned with advanced reasoning abilities; (iii) *math models* that have been augmented with more mathematical skills; and (iv) *domain-specific climate models* that have been continuously pre-trained with climate-relevant corpus. We carefully analyze the evaluation results and summarize the following findings:

- **Finding 1.** *Reasoning models (such as GPT-o3-mini and Deepseek-R1) outperform instruction, math, and domain-specific models, demonstrating the superior significance of advanced reasoning ability in atmospheric science tasks.*
- **Finding 2.** *The inference time scaling introduces interesting quality-efficiency tradeoffs for reasoning models—model accuracy improves with longer reasoning token lengths up to a certain threshold, beyond which the gains plateau and diminishing returns emerge.*
- **Finding 3.** *Despite their superior reasoning capabilities, advanced models remain sensitive to symbolic perturbations—minor changes in variable values or structure can notably impact their accuracy, suggesting that they possibly rely on pattern matching rather than genuine reasoning.*

## 2 Related Work

**LLM advances.** LLMs, such as OPT [14], LLAMA [15], GPT [16], GEMINI [17], CLAUDE [18], and MIXTRAL[19], have demonstrated remarkable performance across a wide range of applications. While general-purpose LLMs exhibit strong adaptability, domain-specific models have also been developed to enhance performance in specialized fields. In the context of atmospheric science, climate-focused LLMs such as CLIMATEBERT [20], and CLIMATEGPT [4] are designed to address the unique challenges of climate modeling and analysis, which illustrates a promising paradigm different from traditional approaches that designing a specific model for some particular task [21, 22, 23, 24, 25]. More recently, reasoning models, including GPT-O1 [26], GEMINI-2.0-FLASH-THINKING [27], QWQ [28], and DEEPSEEK-R1 [29], have emerged, highlighting advancements in mathematical and scientific problem-solving. These models leverage sophisticated reasoning techniques, presenting exciting opportunities for tackling complex challenges in atmospheric science. Details of fundamental differences between reasoning and instruction-tuned models are provided in Appendix B.3.

**LLM benchmarks.** Assessing LLMs is crucial for ensuring their effectiveness in deployment across various domains [30]. Traditional benchmarks such as GSM8K [31] and MATH [12] have become less informative as state-of-the-art models achieve near-perfect scores, motivating the need for more challenging and discriminative benchmarks to evaluate reasoning capabilities. In response, several recent benchmarks have been developed to target more advanced scientific reasoning: GPQA-Diamond [32] focuses on expert-level science, AIME2024 [33] targets advanced mathematical problem solving, and SCIBENCH [13] evaluates collegiate-level scientific reasoning. These benchmarks have been widely adopted for assessing reasoning-oriented LLMs. More recently, a complementary trend has emerged. HUMANITY'S LAST EXAM [34] provides broad subject coverage across the frontier of human knowledge and aims to serve as the final closed-ended academic benchmark of its kind. In contrast, domain-specific efforts such as PHYSICS [35], PhysBench [36], and SciEx [37] offer specialized evaluations in physics and scientific reasoning, incorporating symbolic derivations, open-ended problem solving, and expert-aligned scoring protocols. However, a comprehensive LLM benchmark tailored for atmospheric science remains underrepresented. While CLIMAQA [38] offers a promising first step, it primarily relies on definition-based assessments and lacks the depth required to evaluate complex numerical and symbolic reasoning essential to real-world atmospheric science problems. Designing a meaningful benchmark in this domain requires principled guidance to ensure robust, accurate, and interpretable evaluation. A notable methodological advancement is the use of symbolic extensions in benchmarking, as demonstrated by GSM-Symbolic [39], VarBench [40], and MM-PhyQA, where controlled variation of problem parameters improves robustness and mitigates contamination. These studies reveal that even small perturbations in problem structure can significantly degrade model performance, highlighting persistent fragilities in LLM reasoning and underscoring the need for rigorous benchmarks tailored to domain complexity.

## 3 Dataset and Question Construction

### 3.1 Dual-format question types.

ATMOSSCI-BENCH is designed with a dual-format question structure comprising MCQs and OEQs. These complementary formats enable us to systematically evaluate both the structured reasoning and expressive problem-solving capabilities of LLMs in atmospheric science.

**Multiple-choice questions.** The use of MCQs serves multiple core objectives: enabling automated scoring, supporting controlled symbolic perturbation, and ensuring unambiguous evaluation. Unlike traditional metrics such as exact match, BLEU, or F1 scores — which primarily assess surface-level similarity — MCQs offer fixed, well-defined answer choices. This significantly reduces ambiguity and enables a more precise assessment of logical inference and conceptual comprehension by the model [41]. To maximize the consistency and scalability of our benchmark, we constructed symbolic MCQ templates from source materials and applied rule-based perturbations to systematically explore model sensitivity to input transformations. The full construction pipeline, including template design, numerical instantiation, and distractor generation, is detailed in Section 3.3.

**Open-ended questions.** In parallel, we include open-ended questions to probe model reasoning in a less constrained setting. OEQs are better suited to reveal genuine reasoning capabilities, free from potential biases introduced by distractor. Prior work has noted that MCQs may allow models to bypass reasoning through answer recognition or pattern matching, rather than demonstrating full comprehension of theoretical concepts [34, 36, 42]. However, OEQs pose challenges in evaluation: the output space is unconstrained and lacks a single, universally accepted metric. Recent benchmarks adopt restrictive answer formats to simplify evaluation. For example, `PhysBench` [36] focuses solely on symbolic expressions, while `Humanity's Last Exam` [34] prohibits open-ended answers and instead relies on exact match or extraction-based evaluation. In contrast, we propose a cascade of evaluators specifically tailored for our OEQs. Our OEQs include a mix of quantitative answers involving numerical values with physical units, as well as symbolic expressions. To robustly evaluate this diverse answer space, our evaluators include a quantity evaluator, a symbolic expression evaluator, and an LLM-as-judge evaluator. We detail the full evaluation method in Section 4.2.

## 3.2 Data Source and Pre-processing

To ensure the rigor and relevance of the benchmark, we curated questions from course materials used in atmospheric science-related classes at our university. These sources provide high-quality, well-established content that aligns with the complexity and depth required for evaluating LLMs in this domain. The detailed design principles are provided in Appendix A. We leverage Mathpix OCR [43], a state-of-the-art OCR (Optical Character Recognition) engine, to extract both questions and their corresponding explanations from the collected materials. For multi-part problems or sequential questions where solving one step is necessary to proceed to the next, we consolidated them into single questions to enhance the complexity and depth of reasoning required. This approach preserves the logical progression of problem-solving, ensuring a comprehensive assessment of model capabilities.

**Category distribution.** The benchmark spans five scientific domains commonly addressed in atmospheric science, each chosen for its foundational importance in climate-related education and research. While only two of these — *atmospheric dynamics* and *atmospheric physics*—are formally categorized under atmospheric science, the remaining three — *geophysics*, *hydrology*, and *physical oceanography* — serve as necessary complementary domains. These fields are tightly coupled with atmospheric processes and are frequently integrated into both academic instruction and real-world research. Together, these five categories ensure broad topical coverage and reflect the inherently interdisciplinary nature of atmospheric science:

- **Atmospheric dynamics** focuses on the motion of the atmosphere, including large-scale weather systems, wind patterns, and governing forces of atmospheric circulation.
- **Atmospheric physics** covers physical processes such as radiation, thermodynamics, cloud formation, and energy transfer within the atmosphere.
- **Geophysics** encompasses the physical processes of the Earth, including its magnetic and gravitational fields, seismic activity, and internal structure.
- **Hydrology** examines the distribution, movement, and properties of water on Earth, including the water cycle, precipitation, rivers, lakes, and groundwater dynamics.
- **Physical oceanography** investigates the physical properties and dynamics of ocean water, including currents, waves, tides, and ocean-atmosphere interactions.

We summarize the distribution of questions across core scientific domains in Table 1, encompassing both MCQs and OEQs in our benchmark. The `MCQ10` subset is constructed by symbolically expanding each of our 67 curated question templates into 10 unique instances (Section 3.3), resulting in 670 scientifically grounded MCQs that enable large-scale, diverse evaluation. To complement this core set, we introduce an additional subset, `MCQ10_EXT`, consisting of 240 questions drawn from

three complementary domains that are closely integrated with atmospheric science. For robustness evaluation, we also construct `MCQ30`, an augmented set incorporating higher levels of symbolic variation, for which 30 unique instances are generated for every question template (Section 5.3). To ensure consistency, `MCQ10_EXT` and `MCQ30` are generated following the same framework as `MCQ10`. In contrast, the `OEQ` dataset (391 questions) is individually curated to emphasize deep reasoning, without symbolic perturbations.

Table 1: Question type counts across different subfields in terms of MCQs and OEQs.

| Type | Atmos. Dynamics | Atmos. Physics | Geophysics | Hydrology | Phys. Oceanog. | Total |
|---|---|---|---|---|---|---|
| MCQ10 | 370 | 140 | 70 | 50 | 40 | **670** |
| MCQ10_EXT | 0 | 0 | 10 | 170 | 60 | **240** |
| OEQ | 46 | 85 | 11 | 226 | 23 | **391** |

## 3.3 MCQ Generation Framework

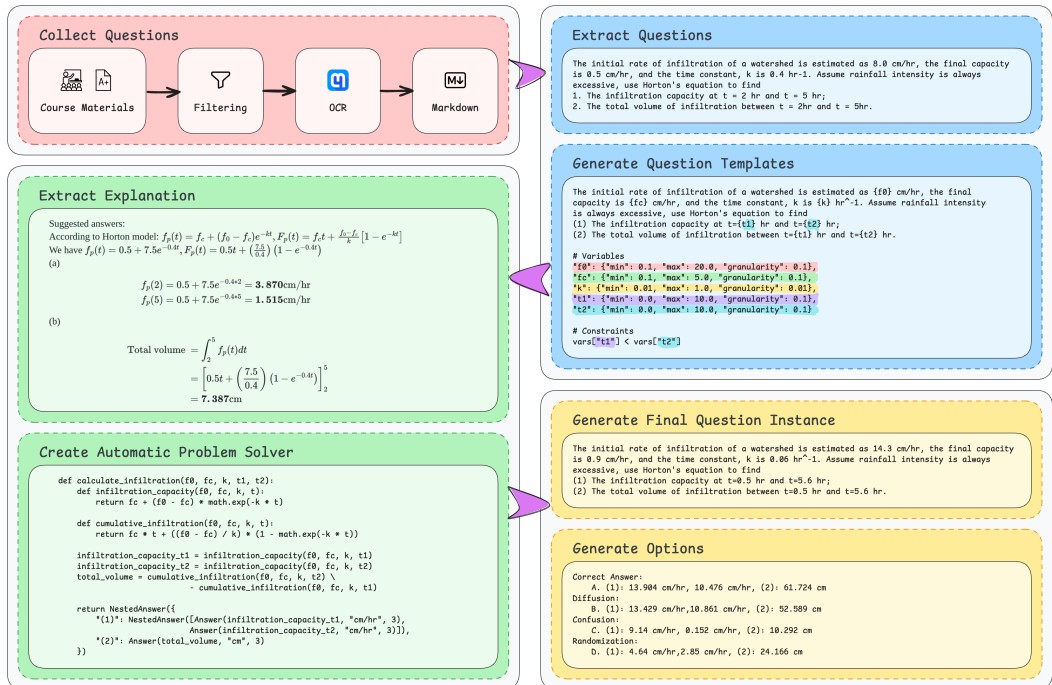

Figure 2: Construction pipeline of our template-based question generation framework. Red block display the question collecting process. Blue blocks represent the question generation process (variables are highlighted in different colors). Greed blocks depict the automatic problem solver, which derives the answer from given variables. Yellow blocks illustrate an example of a generated question and its corresponding options.

To rigorously evaluate the reasoning and problem-solving capabilities of LLMs, we employ symbolic MCQ generation techniques inspired by the `GSM-Symbolic` framework [39], enhanced with a rule-based mechanism. This approach enables the creation of scalable and diverse question sets while ensuring logical coherence and alignment with real-world physical laws. Instead of fixed numerical values, we also design a template-based question perturbation mechanism with placeholder variables, which can be systematically instantiated through symbolic extensions. This ensures that models are tested on genuine reasoning ability rather than pattern matching from the potentially contaminated training data. Figure 2 illustrates the question construction pipeline as we enumerate below.

- **Question template construction**: We invite domain experts in atmospheric science to systematically transform selected questions (OCR extracted) into reusable templates. The experts manually identify numerical values within each question and replace them with variable placeholders, ensuring flexibility for symbolic instantiation. These variable placeholders, highlighted in different

colors in Figure 2, allow for systematic variation while preserving the original scientific integrity of the problem.

- **Numerical assignment in question template**: We design a rule-based mechanism for valid numerical assignments in each question template. Note that many variables in atmospheric science problems are interdependent, meaning that the inappropriate assignment of some value(s) could lead to unrealistic or invalid physical scenarios. To fulfill this requirement, we ask the experts for each question template to define: (i) a valid numerical range (*min*, *max*) for each variable to ensure scientifically plausible values; (ii) a granularity parameter (i.e., the smallest step size between values) to control precision; and (iii) a set of rule-based constraints that are manually implemented to enforce logical dependencies (e.g., in Figure 2, ensuring $t_1 < t_2$). We believe these manual configurations ensure that all generated instances remain scientifically valid while allowing systematic variation in numerical representation.

- **Automatic problem solver to support value perturbation**: For each question, we utilize GPT-4o to generate an initial Python implementation based on the corresponding explanatory solution. This synthesized solution is then *manually reviewed, verified, and refined by experts to ensure correctness* and adherence to the intended problem-solving methodology. Once validated, the solver can automatically compute the correct answer for any given set of valid input variables, ensuring consistency and scalability in question generation. Note that to ensure consistency, accuracy, and alignment with real-world scientific standards, we also manually assign appropriate units and define significant digits for rounding the final answer in each automatic problem solver. This standardization maintains numerical precision while preventing inconsistencies in representation, ensuring that generated answers adhere to established atmospheric science conventions.

- **Option generation**: To effectively assess LLM reasoning, MCQs require plausible but incorrect distracting options that challenge the model's understanding while avoiding trivial elimination strategies [44]. To mitigate potential positional bias, we randomized the order of options during benchmark generation. We further verified that model accuracy was not skewed toward any specific answer position. The detailed generation mechanisms are provided in Appendix F.

## 4 Evaluation and Experimental Setup

### 4.1 Constrained Prompting for Evaluation

To ensure consistency in model outputs and simplify downstream answer extraction, we adopt *constrained prompting* across both MCQ and OEQ tasks. All prompts are designed to guide the model toward producing structured answers, with final results required to appear within LaTeX \boxed{} expressions. These outputs are subsequently extracted using regular expression (RegEx) parsing to support reliable and automatic evaluation. Full prompt templates are provided in Appendix E.1.

### 4.2 Evaluation Metrics

We design separate evaluation protocols for multiple-choice and open-ended questions, aligned with the structure of their respective answer formats.

**MCQ Evaluation.** For multiple-choice questions, we use the `MCQEvaluator`, which applies a straightforward accuracy-based metric. We extract the model's selected option from its response using a regular expression and compare it against the ground-truth label. A match is counted as correct; otherwise, it is marked incorrect. The final metric is the proportion of correctly matched answers across the evaluation set.

**OEQ Evaluation.** For open-ended questions, which can yield answers in numeric, symbolic, or natural language formats, we adopt a cascade of evaluators to improve coverage and accuracy:

- **QuantityEvaluator:** This is the primary evaluator for arithmetic-based questions that produce scalar answers with physical units. We use regular expressions to extract numeric values from model outputs and validate correctness within a *5% tolerance*, which reflects standard error bounds in Earth Science disciplines. Unit compatibility and conversion are handled using the `pint` Python library.

- **ExpressionEvaluator:** If the quantity-based check fails or the format is not purely numeric, we fall back to symbolic equivalence checking. We parse both model and reference expressions using the Python library `sympy` and simplify their difference. This approach is inspired by methodologies in recent scientific reasoning benchmarks such as *PHYSICS* [35].

- **LLMEvaluator:** If both automated evaluators fail, we invoke an GPT-4O-MINI [16] with structured outputs [45] to compare responses against the ground truth using rubric-based instructions (Similar to QuantityEvaluator, *5% tolerance* for numerical value are accepted). An example LLMEvaluator prompt is provided in Appendix E.2. Notably, LLM-based evaluation is increasingly accepted in scientific benchmarks. Recent studies have shown that LLMs can serve as effective graders, exhibiting strong agreement with expert human evaluations [37]. Several state-of-the-art benchmarks—including *SciEx* [37], *Humanity's Last Exam* [34], and *PHYSICS* [35]—have adopted LLM-as-Judge as a primary or even exclusive evaluation method.

If any of the above evaluators determine the model's output to be equivalent to the reference, the sub-question is marked as correct. The final score for an OEQ is computed as the average correctness across all its subparts, and the overall OEQ accuracy is the average across the dataset. While `QuantityEvaluator` and `ExpressionEvaluator` are highly accurate and well-grounded for positive true answers, they may still produce false negatives due to the evaluation in Python library `sympy`. Mitigating false negatives is one of the design intentions of our cascade of evaluators, which `LLMEvaluator` plays as a solid backup evaluator handling potential positive cases that `ExpressionEvaluator` marks as false, thereby reinforcing overall robustness. Further analysis of evaluator robustness, the human-LLM agreement study on `LLMEvaluator`, and false negatives examples occur in `ExpressionEvaluator` are provided in Appendix J.

### 4.3 Evaluation Questions

We design four main experiments to assess LLM performance on our benchmark, focusing on comprehensive performance comparison across model categories (*Q1*), the effect of inference-time reasoning length (*Q2*), and robustness to symbolic perturbation (*Q3*). We enumerate these concrete questions below:

- *Q1. How do various state-of-the-art LLMs (i.e., falling into different categories of instruction, math, reasoning, and domain-specific models) comprehensively perform for the proposed atmospheric science benchmark?*
- *Q2. How do the models specialized in reasoning perform during inference time scaling, i.e., how can we improve the model's test accuracy by increasing the length of reasoning tokens?*
- *Q3. How robust are the benchmark results, especially when we variate the degree of perturbation introduced by symbolic variation?*

### 4.4 Benchmark Models

To comprehensively assess LLM performance in atmospheric science, we evaluate a diverse set of state-of-the-art models spanning four categories: (i) instruction-tuned models, (ii) reasoning-optimized models, (iii) math-augmented models, and (iv) domain-specific models. This categorization facilitates a structured comparison between general-purpose, specialized, and domain-adapted approaches. A complete list of evaluated models is provided in Appendix B.1, and the corresponding cost and runtime statistics are reported in Appendix M.

## 5 Evaluation Results and Discussion

### 5.1 End-to-end Evaluation Results

**Experimental setup**. To comprehensively evaluate the performance of four categories of LLMs on atmospheric science tasks and assess whether ATMOSSCI-BENCH provides a sufficiently challenging and discriminative evaluation framework, we conduct a systematic performance comparison using our `MCQ10`, `MCQ10_EXT`, and `OED` benchmark across four representative LLM categories introduced in Section 4. We standardize experimental settings for each category as: (i) Reasoning models use 32K max context length, including the reasoning tokens; (ii) Instruction and math models use 8K max output tokens, balancing response quality and efficiency; (iii) Domain-specific models are set to 4K context length, the maximum capacity they support. By controlling these variables, we ensure that performance differences reflect genuine capability gaps rather than confounding factors, allowing us to validate whether ATMOSSCI-BENCH effectively differentiates model performance and highlights reasoning proficiency. Details of the hyperparameter settings and our computation resource are provided in Appendix D and Appendix G.

**Results and analysis**. For `MCQ10`, we present accuracy across different atmospheric science tasks, along with an overall performance comparison in Table 2 with three key observations:

Table 2: Accuracy (%) and symbolic standard deviation (SymStd.) comparison across four LLM categories on the `MCQ10` dataset.

| Category | Model | Hydro | AtmDyn | AtmosPhy | GeoPhy | PhyOcean | Overall Acc | SymStd. |
|---|---|---|---|---|---|---|---|---|
| **Instruction Models** | Gemma-2-9B-it | 28.0 | 17.29 | 21.42 | 11.42 | 20.0 | 18.50 | 3.73 |
| | Gemma-2-27B-it | 56.0 | 31.08 | 47.14 | 41.42 | 40.0 | 37.91 | 4.62 |
| | Qwen2.5-3B-Instruct | 46.0 | 29.19 | 34.28 | 30.0 | 37.5 | 31.49 | 7.71 |
| | Qwen2.5-7B-Instruct | 60.00 | 38.11 | 50.71 | 51.43 | 32.50 | 43.43 | 4.90 |
| | Qwen2.5-32B-Instruct | 60.00 | 46.22 | 63.57 | 62.86 | 50.0 | 52.84 | 5.68 |
| | Qwen2.5-72B-Instruct-Turbo | 72.00 | 50.00 | 76.43 | 44.29 | 57.50 | 57.01 | 4.44 |
| | Llama-3.3-70B-Instruct | 82.0 | 42.66 | 66.43 | 51.52 | 42.5 | 51.51 | 3.92 |
| | Llama-3.1-405B-Instruct-Turbo | 70.00 | 48.11 | 64.29 | 57.14 | 52.50 | 54.33 | 5.81 |
| | GPT-4o-mini | 48.00 | 42.16 | 58.57 | 40.00 | 40.00 | 45.67 | 5.08 |
| | GPT-4o | 72.0 | 51.35 | 74.29 | 60.0 | 45.0 | 58.21 | 5.22 |
| | Gemini-2.0-Flash-Exp | 90.00 | 58.11 | 67.14 | 77.14 | 55.00 | 64.18 | 3.85 |
| | Deepseek-V3 | 94.00 | 56.22 | 73.57 | 64.29 | 52.50 | 63.28 | 6.02 |
| **Reasoning Models** | QwQ-32B-Preview | 88.0 | 63.24 | 87.86 | 77.14 | 50.0 | 70.9 | 4.41 |
| | Gemini-2.0-Flash-Thinking-Exp (01-21) | 100.00 | 78.11 | 83.57 | 91.43 | 70.00 | 81.79 | 3.78 |
| | GPT-o1 | 100.00 | 82.70 | 90.71 | 92.86 | 77.50 | 86.42 | 2.94 |
| | Deepseek-R1 | 98.00 | 85.68 | 93.57 | 95.71 | 72.50 | 88.51 | 2.99 |
| | Qwen3-235B-A22B-FP8-Throughput | 98.0 | 86.49 | 93.57 | 90.0 | 70.0 | 88.21 | 3.75 |
| | GPT-o3-mini | 100.0 | 87.57 | 89.29 | 95.71 | 77.5 | 89.1 | 3.3 |
| **Math Models** | Deepseek-Math-7B-RL | 22.00 | 22.43 | 28.57 | 24.29 | 35.00 | 24.63 | 4.52 |
| | Deepseek-Math-7B-Instruct | 36.00 | 28.38 | 33.57 | 30.00 | 40.00 | 30.90 | 4.17 |
| | Qwen2.5-Math-1.5B-Instruct | 48.00 | 29.19 | 23.57 | 34.29 | 30.00 | 30.00 | 2.94 |
| | Qwen2.5-Math-7B-Instruct | 54.00 | 30.81 | 39.29 | 35.71 | 30.00 | 34.78 | 5.85 |
| | Qwen2.5-Math-72B-Instruct | 68.00 | 54.05 | 72.14 | 62.86 | 30.00 | 58.36 | 6.31 |
| **Domain-Specific Models** | ClimateGPT-7B | 26.00 | 18.65 | 22.86 | 11.43 | 32.50 | 20.15 | 5.18 |
| | ClimateGPT-70B | 24.00 | 25.41 | 28.57 | 40.00 | 22.50 | 27.31 | 4.45 |
| | GeoGPT-Qwen2.5-72B | 60.0 | 30.0 | 40.71 | 37.14 | 25.0 | 34.93 | 2.74 |

- ATMOSSCI-BENCH *effectively differentiates LLM performance across categories, with reasoning models demonstrating the highest proficiency.* The results confirm that our benchmark successfully distinguishes LLM performance, particularly in assessing reasoning proficiency. Reasoning models (70.9% - 89.1%) significantly outperform instruction models (18.5% - 64.18%), demonstrating superior consistency with lower symbolic reasoning standard deviation (SymStd) [39]. GPT-O3-MINI, the best-performing reasoning model, achieves 89.1% accuracy, while the top instruction model, GEMINI-2.0-FLASH-EXP, only reaches 64.18%, a substantial 24.92% gap. This clear performance variance underscores ATMOSSCI-BENCH 's ability to challenge advanced LLMs, ensuring that strong reasoning skills translate into measurable performance gains.

- *Math models do not show a clear advantage over instruction models.* Despite their specialization, math models do not significantly outperform instruction models, suggesting that mathematical optimization alone is insufficient for solving atmospheric science challenges.

- *Domain-specific models underperform despite climate specialization, indicating a need for strong reasoning-augmented approaches.* Domain-specific models show notably lower performance despite being trained on domain-related corpora. For instance, CLIMATEGPT-7B and CLIMATEGPT-70B achieve only 20.15% and 27.31% accuracy, respectively. This outcome is not entirely surprising, as the pretraining data for ClimateGPT is drawn almost entirely from crowd-sourced prompt-completion pairs and multi-turn dialogues, which are not effective for structured, step-wise reasoning found in scholarly problem solving. Interestingly, GEOGPT-QWEN2.5-72B also underperforms, achieving only 34.93% accuracy, whereas its base model Qwen-2.5-72B-Instruct-Turbo with the same model size achieves an accuracy of 57.01%. To better understand the flaws of domain-specific models, we conducted an error analysis, as detailed in Appendix I.2. These error patterns are consistent with our explanation of GeoGPT's. Indeed, GeoGPT's stated design goal: it is intended "to understand the demands of non-professional users, then think, plan, and execute defined GIS tools to produce effective results." Optimizing for tool planning rather than sustained step-by-step problem-solving data, therefore, provides a more plausible explanation for its lower accuracy than parameter size. Taken together, the evidence suggests that the principal limitations of current domain-specific LLMs lie in their training objectives and data composition: a focus on tool interaction or conversational style can leave gaps in both domain knowledge breadth and reasoning robustness. This underscores the need for reasoning-augmented domain models that combine scientific knowledge with structured inference skills. ATMOSSCI-BENCH provides a rigorous and interpretable evaluation framework to support the development and diagnosis of such models, addressing key limitations in current domain-specific approaches.

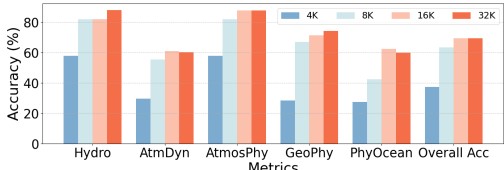

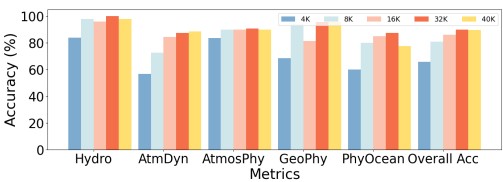

(a) QwQ-32B-Preview accuracy over 4K–32K tokens.  (b) GPT-o3-mini accuracy over 4K–40K tokens.

Figure 3: Reasoning step study. Accuracy (%) of different models across increasing input lengths.

The results for `OEQ` and the complementary `MCQ10_EXT` set are provided in Appendix H. `MCQ10_EXT`, designed to augment the original `MCQ10` set with additional domain coverage, exhibits performance trends consistent with those observed in `MCQ10`, supporting the robustness of our findings. In contrast, LLMs score below 40% on the `OEQ`, highlighting ATMOSSCI-BENCH's capacity to evaluate deeper layers of logical and numerical reasoning, reinforcing the importance of inference-through-thinking as a critical dimension in assessing LLM capabilities for scientific problem-solving.

In conclusion, to answer *Q1* regarding the overall performance of various LLM categories, our evaluation reveals that *reasoning models significantly outperform instruction, math, and domain-specific models in atmospheric science tasks, highlighting their superior adaptability to advanced reasoning challenges, while domain-specific models struggle despite specialized training.*

### 5.2  Inference Time Scaling for Reasoning Models

**Experimental setup**. To answer *Q2*, i.e., whether increasing the length of reasoning tokens improves the performance of reasoning models, we conduct an inference time scaling evaluation on `MCQ10` using the QWQ-32B-PREVIEW and GPT-O3-MINI model, varying its reasoning token limits from 4K up to 40K. By systematically increasing the token limit, we aim to determine whether a longer inference process leads to higher accuracy and whether there exists an optimal threshold beyond which additional tokens provide minimal benefit.

**Results and analysis**. As shown in Figure 3a, increasing the reasoning token limit for QWQ-32B-PREVIEW generally improves model accuracy, but the gains diminish beyond a certain threshold. The performance in terms of overall accuracy is consistently lower at 4K tokens, improves significantly at 8K and 16K tokens, and then plateaus beyond 16K tokens, with 32K tokens offering only marginal improvement. GPT-O3-MINI in 3b show a similar pattern except plateaus around 32K. This trend suggests that while extending reasoning length enhances model performance up to a certain point, it further increases yield, diminishing returns without proportional accuracy gains. Thus, our answer to *Q2* is that *increasing the length of reasoning tokens improves model accuracy up to 16K tokens, beyond which performance gains diminish, indicating an optimal threshold for inference time scaling.*

### 5.3  Robustness of ATMOSSCI-BENCH

To evaluate the robustness of ATMOSSCI-BENCH (*Q3*), we conduct a experiment to assess - robustness to different degrees of perturbation introduced by symbolic variation.

**Experimental setup**. Inspired by `GSM-Symbolic` [39], which demonstrates that modifying numerical variables in the `GSM8K` dataset leads to significant performance drops, suggesting that LLMs may rely on pattern matching rather than genuine logical reasoning. Here, pattern matching refers to reliance on patterns memorized during training—such as recalling similar questions or solutions—rather than reasoning through the problem independently. We aim to assess the robustness of advanced reasoning models under varying degrees of symbolic perturbation. To examine this, we evaluate three reasoning models—DEEPSEEK-R1, GEMINI-2.0-FLASH-THINKING-EXP (01-21), GPT-O3-MINI, and QWQ-32B-PREVIEW—on `MCQ30`, which consists of 30 test sets for each question template, with controlled symbolic variations to analyze sensitivity to numerical perturbations. We systematically modify numerical variables within a scientifically reasonable range, introducing controlled variations to assess whether performance remains stable or degrades significantly with perturbation.

**Results and analysis**. Figure 4 illustrates the empirical performance distribution of reasoning models on `MCQ30`. We observe that the accuracy of the original question set for all models (dashed line in Figure 4) is approximately one standard deviation away from the mean accuracy across perturbed instances, except QWQ-32B-PREVIEW exhibits about two standard deviations. A hypothesis test was also conducted (Appendix N), which indicates that the null hypothesis of "random chance" cannot be conclusively rejected, suggesting the possibility of potential data contamination. Although not

definitive evidence of contamination, such deviations are nontrivial and warrant further attention, implying that reasoning LLMs are possibly pattern-matching from contaminated data. To answer *Q3* w.r.t symbolic variation, the results indicate that *the reasoning models evaluated in our benchmark could still be under the risk of insufficient robustness under symbolic perturbation, as increasing the degree of variation leads to notable and often unpredictable drops in accuracy, suggesting that they possibly rely on pattern matching instead of genuine reasoning.*

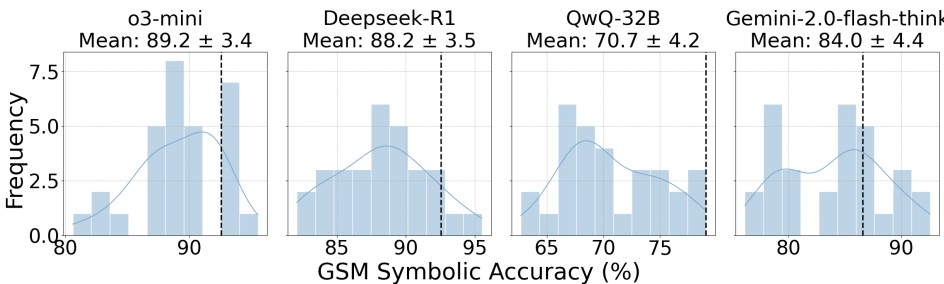

Figure 4: Performance distribution among reasoning LLMs on `MCQ30`. The Y-axis represents the frequency of the symbolic test sets achieving the accuracy shown on the X-axis. The black vertical dash lines denote the accuracy of the original question set.

## 5.4 Further Discussion

To better understand the skill-level limitations of current LLMs in atmospheric science, as well as common error patterns and evaluation constraints, we refer readers to Appendix K (skill-oriented ablation analysis including a retrieval-augmented generation (RAG) baseline), Appendix I.1 (error categorization), Appendix N (data contamination), and Appendix O (discussion of limitations).

## 6 Conclusion

In this paper, we introduced ATMOSSCI-BENCH, a novel benchmark designed to systematically evaluate the reasoning and problem-solving capabilities of LLMs in atmospheric science. Our findings highlight that reasoning models outperform other categories, demonstrating stronger problem-solving and reasoning capabilities in the domain of atmospheric science. This also underscores the benchmark's effectiveness in differentiating models. Our benchmark covers five core categories — hydrology, atmospheric dynamics, atmospheric physics, geophysics, and physical oceanography — through a dual-format question design comprising both MCQs and OEQs. This structure enables both scalable automated evaluation and deeper probing of scientific reasoning skills. ATMOSSCI-BENCH employs a scalable, template-based generation framework for MCQs to ensure diversity and control over symbolic perturbations, while OEQs are curated to assess deeper reasoning skills without predefined answer choices. Through a comprehensive evaluation across four model categories — instruction-tuned models, advanced reasoning models, math-augmented models, and domain-specific climate models — we provide key insights into the strengths and limitations of current LLMs. Our findings highlight that reasoning models outperform other categories, demonstrating stronger problem-solving and reasoning capabilities in the domain of atmospheric science. This also underscores the benchmark's effectiveness in differentiating models. We believe that ATMOSSCI-BENCH (where all the implementations are fully open-sourced) can serve as an essential step toward advancing the application of LLMs in climate-related decision-making by offering a standardized and rigorous evaluation framework for future research.

## Acknowledgments and Disclosure of Funding

This research is supported by the Hong Kong Research Grants Council's collaborative research funds (project nos. C6032-21G, C5004-23G, C5085-24G), general research fund (project no. 16300424), theme-based research scheme (project no. T22-501/23R, T22-607/24-N) and NSFC/RGC Collaborative Fund (project no. CRS_PolyU503/23). All co-authors acknowledge the support by the Otto Poon Centre for Climate Resilience and Sustainability at HKUST. This work is part of the United Nations Educational Scientific and Cultural Organization (UNESCO) International Decade of Sciences for Sustainable Development (2024–33) and contributes to the Seamless Prediction and Services for Sustainable Natural and Built Environment (SEPRESS) Program (2025–32), an initiative endorsed under this global framework.

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

# A   Questions design principles.

To ensure a rigorous evaluation of LLMs in atmospheric science, we adhere to a set of well-defined principles that emphasize reasoning and interpretative abilities:

**Deep understanding of essential physical equations:**   Atmospheric science is governed by fundamental physical equations, and a meaningful evaluation requires that LLMs not only recall these principles but also apply them appropriately in the corresponding contexts. Thus, the questions should be designed to assess both conceptual comprehension and the ability to use these equations in problem-solving, ensuring the benchmark measures true scientific reasoning rather than mere memorization.

**Complex reasoning and multi-step logic:**   Many real-world atmospheric problems require synthesizing information from multiple sources, integrating equations, and applying multi-step logical reasoning. To reflect these challenges, benchmark questions should be crafted to go beyond simple recall, testing the model's ability to handle intricate reasoning and dynamic problem-solving scenarios inherent to the field.

**Appropriate numerical arithmetic processing:**   Accurate numerical computation is essential for scientific disciplines, where correct reasoning leads to fixed, verifiable answers. By incorporating numerical problems, we provide a structured and objective evaluation framework, eliminating ambiguities in assessment. This approach also enables seamless integration of reasoning tasks, extending the benchmark's scope to evaluate mathematical intuition and computational fluency.

# B   Model Usage

## B.1   Model List

**Instruction models**. Instruction-tuned models serve as strong general-purpose baselines, optimized for following prompts and single-step inference tasks, where we include:

- GPT-4O, GPT-4O-MINI [16]: OpenAI's instruction-tuned models.
- QWEN2.5-INSTRUCT (3B, 7B, 32B, 72B) [46]: Instruction-tuned Qwen models with enhanced abilities.
- GEMMA-2-9B-IT, GEMMA-2-27B-it [47]: Google's open-weight instruction models; along with Gemini-2.0-Flash-Exp [48], the powerful Gemini model optimized for efficiency.
- LLAMA-3.3-70B-INSTRUCT, LLAMA-3.1-405B-INSTRUCT-TURBO [49]: Meta's widely used instruction models.
- DEEPSEEK-V3 [50]: Deepseek's latest MoE-based instruction model for general tasks.

**Math models**. Mathematical LLMs specialize in problem-solving, computational reasoning, and theorem proving — such ability is essential for atmospheric problems. Towards this end, we include:

- DEEPSEEK-MATH-7B-INSTRUCT and DEEPSEEK-MATH-7B-RL [51]: Deepseek's math-focused models trained for theorem proving.
- QWEN2.5-MATH (1.5B, 7B, 72B) [52]: Qwen's recent models optimized for mathematics.

**Reasoning models**. Reasoning ability is the core technique to improve LLMs' performance over complicated tasks. We include the recent advanced reasoning models focus on deep logical reasoning and multi-step problem-solving:
- GPT-O1 [26], GPT-O3-MINI [53]: OpenAI's reasoning-optimized model.
- QWQ-32B-PREVIEW [28]: Reasoning model based on Qwen2.5-32B.
- GEMINI-2.0-FLASH-THINKING-EXP (01-21) [27]: Extended Gemini-2.0-Flash-Exp for enhanced reasoning.
- DEEPSEEK-R1 [29]: Deepseek's RL-trained model for complex problem-solving.
- QWEN3-235B-A22B-FP8-THROUGHPUT [54]: The Latest generation in the Qwen series equipped with thinking mode to enhance its reasoning capabilities.

**Domain-specific models**. We also include some models that are specially tailored for climate-related and atmospheric science tasks by supervised fine-tuning or continuous pre-training:

- CLIMATEGPT-7B, CLIMATEGPT-70B [4]: QA models specialized in the climate domain.
- GEOGPT-QWEN2.5-72B [55]: A domain-adapted model built on Qwen2.5-72B supports advanced reasoning and knowledge synthesis in specialized areas of geoscience.

## B.2 Model and Library Usage Licenses

We list in Table 3 the models and software assets used in this work, along with their respective sources and licensing terms. All API-based models are accessed through official platforms under standard usage policies, while open-source models are released under community-accepted licenses (Apache, MIT, etc.).

| Model / Library | Source / Access Method | License / Terms of Use |
|---|---|---|
| GPT-4o, GPT-4o-mini, GPT-o1, GPT-o3-mini | OpenAI API | OpenAI API Terms of Use |
| Gemini-2.0-Flash-Exp, Gemini-2.0-Flash-Thinking-Exp (01-21) | Google API (Vertex AI) | Google Cloud Terms of Service |
| DeepSeek-V3, DeepSeek-R1, DeepSeek-Math-7B-Instruct, DeepSeek-Math-7B-RL | Deepseek API | Deepseek Public API Terms |
| Qwen2.5-Instruct (3B, 7B, 32B, 72B) | HuggingFace / Together AI | Apache License 2.0 |
| Qwen2.5-Math (1.5B, 7B, 72B) | HuggingFace / Together AI | Apache License 2.0 |
| QwQ-32B-Preview | HuggingFace (based on Qwen2.5-32B) | Apache License 2.0 |
| Qwen3-235B-A22B-FP8-Throughput | HuggingFace | Apache License 2.0 |
| Gemma-2-9B-it, Gemma-2-27B-it | HuggingFace (Google) | CC BY-NC 4.0 / Google Research Terms |
| Llama-3.3-70B-Instruct, Llama-3.1-405B-Instruct-Turbo | HuggingFace (Meta AI) | Meta Llama 3 Community License Agreement |
| ClimateGPT, GeoGPT | HuggingFace | License provided in original repo (research only) |
| HuggingFace Transformers | `https://github.com/huggingface/transformers` | Apache License 2.0 |
| Accelerate | `https://github.com/huggingface/accelerate` | Apache License 2.0 |
| Ray | `https://github.com/ray-project/ray` | Apache License 2.0 |
| NumPy, SciPy, Pandas | PyPI / open-source | BSD / MIT Licenses |

Table 3: Sources and license information for models and libraries used in this work.

## B.3 Fundamental Differences Between Reasoning and Instruction-Tuned Models

We provide a deeper explanation from a technical and architectural perspective regarding the **fundamental differences** that separate "reasoning models" (e.g., GPT-O1) from "instruction-tuned models" (e.g., GPT-4O):

1. **Deliberate "Thinking" Stage + Extra Inference Compute:** Reasoning models insert a dedicated "<think>" phase and allocate additional compute at inference time, enabling long chain-of-thought processing that allows smaller reasoning models to outperform much larger instruction-tuned ones on complex tasks [26, 56, 29].

2. **Self-Checking to Limit Error Propagation:** Through reinforcement learning, models develop emergent behaviors—self-verification, reflection, and back-tracking—that catch and correct faulty reasoning steps before the final answer [26, 29].

3. **Specialized RL Post-Training for Reasoning:** Models like DEEPSEEK-R1 rely on pure RL fine-tuning with *Group Relative Policy Optimization (GRPO)* on step-by-step solution data, explicitly rewarding correct intermediate reasoning and strengthening overall chain-of-thought quality [29, 28].

For these reasons, reasoning models show significant improvements in complex **multi-step** reasoning tasks. As shown on line 88, all reasoning models achieve substantial gains on benchmarks tailored for such tasks [26, 56, 28].

ATMOSSCI-BENCH is specifically designed to evaluate whether LLMs equipped with thinking stages, intermediate learning, and self-correction can effectively solve multi-step problems. The results

confirm that ATMOSSCI-BENCH successfully distinguishes reasoning models from instruction-tuned models in atmospheric science, thus demonstrating its ability to assess reasoning capacity while posing meaningful challenges for future LLM development.

## C   Data Source and Usage Statement

The benchmark dataset introduced in this paper was independently constructed by the authors. All questions and materials were derived and reformulated from our available university-level content in atmospheric science-related courses. These materials include lecture notes, problem sets, and instructional examples used for teaching at our institution.

No proprietary, copyrighted, or scraped content was included in the dataset. The resulting benchmark is intended solely for academic research and educational use. We confirm that the dataset does not contain any personal information, and sensitive data that may negatively impact society.

To the best of our knowledge, this benchmark complies with relevant institutional and academic usage policies, and poses no legal or ethical risk for public release.

## D   Hyperparameters

To ensure fair comparison and consistent evaluation, we standardize the inference-time hyperparameters across all models in accordance with their capabilities and design constraints.

For **reasoning-optimized models**, we use a maximum context length of **32K tokens**. This decision is motivated by the fact that DEEPSEEK-R1 has a fixed 32K context window that cannot be modified. To maintain fairness, we adopt the same 32K limit for all reasoning models, including GPT-O1, QwQ-32B-PREVIEW, and GEMINI-2.0-FLASH-THINKING-EXP (01-21). This configuration provides sufficient space for long-form reasoning and multi-step inference, ensuring that reasoning performance is not artificially constrained by token limits.

Additionally, most reasoning models—such as GPT-O1 and DEEPSEEK-R1—do not support customized decoding parameters like `temperature`, `top_p`, or `repetition_penalty`. Therefore, we use default hyperparameters for all models across all categories to ensure evaluation consistency and reproducibility.

For **instruction-tuned**, **math-augmented**, and **domain-specific models**, we set the maximum token limit to **8K**, which provides ample context for solving our benchmark tasks given the typical response lengths of these models.

## E   Prompt

### E.1   LLM Prompt Template

We include below the full prompt templates used for both MCQ and OEQ questions. These were passed directly to the language models to elicit structured, parseable responses.

**MCQ prompting.**   For multiple-choice questions, prompts are structured to elicit step-by-step reasoning followed by a clearly formatted answer selection. The model is instructed to return the final choice using the exact format \boxed{A/B/C/D}, ensuring compatibility with our extraction script.

---

**MCQ Prompt Template**

```
You are an Earth Science Expert answering multiple-choice questions.
Here is the question:  {question}
Here are the options:
{options_str}

Instructions:
1. Carefully analyze the question and options provided.
2. Please think step by step.  Use logical reasoning and critical thinking to
generate a detailed explanation or steps leading to the answer.
3. At the end of your response, ensure to provide the correct option
(A/B/C/D) on a new line in the following format strictly:
**Final Answer**:  \[ \boxed{{A/B/C/D}} \]
```

---

**OEQ prompting.** For open-ended questions, prompts instruct the model to produce a full derivation or explanation, concluding with boxed answers. When questions include multiple subparts (e.g., a), b)), each should be addressed in order, with the corresponding boxed result. This constrained prompting strategy enhances interpretability, ensures evaluation robustness, and minimizes ambiguity in final output formatting.

---

**OEQ Prompt Template**

```
You are an expert in Earth System Science.  Think step by step using logical
reasoning and scientific principles.
Provide a detailed explanation or derivation leading to your answer.
If the question includes subparts (e.g., a), b)), address each subpart
sequentially.
Conclude each subpart with its final result formatted as a LaTeX expression,
using:
a) \boxed{...}
b) \boxed{...}
For single-part questions, conclude with a single \boxed{final_answer}.
```

---

## E.2   LLM Evaluator Prompt

OpenAI provides support for structured outputs [45], enabling model responses to adhere to predefined JSON schemas. Following this approach, we define the following `AnswerResponse` schema using `pydantic` to strictly constrain the LLM's output format:

---

**AnswerResponse Schema (Pydantic)**

```
from pydantic import BaseModel
class AnswerResponse(BaseModel):
  is_correct:  bool
  explanation:  str
```

---

The prompt below is used to guide the LLM evaluator to assess answer correctness based on mathematical, physical, and conceptual equivalence:

---

**LLMEvaluator Prompt**

```
You are an expert physics teacher evaluating student answers.
Compare the following two answers and determine if they are equivalent.

Consider the following in your evaluation:
1. Mathematical equivalence (e.g., 2π = 6.28)
2. Physical unit equivalence (e.g., 1 m/s = 3.6 km/h)
3. Conceptual equivalence (e.g., F = ma and a = F/m)
4. Numerical tolerance:  Allow a tolerance of {self.tolerance * 100}% for
numerical values.
   (e.g., if the expected value is 10, values between {10 - 10 *
self.tolerance} and {10 + 10 * self.tolerance} are acceptable.)

Respond with is_correct (true/false) and an explanation.
Expected answer (in LaTeX): {expected}
Student answer (in LaTeX): {actual}
```

---

## F   Incorrect option generation for MCQ Generation

We design the following mechanisms to generate incorrect options: (i) Diffusion: producing an incorrect answer by randomly swapping two variables in the computation; (ii) Confusion: altering a single variable in the equation to generate a close but incorrect result; (iii) Randomization: randomly assigning all variables within their predefined constraints, ensuring adherence to the rule-based mechanism; and (iv) Default: if above three methods fail to generate valid incorrect options (i.e., those satisfying the scientific constraints of the rule-based mechanism), we use a default strategy, where incorrect options are generated as scaled multiples of the correct answer (e.g., $\times 2, \times 3, \times 4$).

## G    Experimental Compute Resources

We categorize LLM inference into two groups based on deployment method: **API-based** and **local-based**.

- **API-based**: Models in this category are hosted by providers such as OpenAI, Google, Deepseek, and TogetherAI. We access these models via public inference APIs. To accelerate large-scale evaluation, we utilize parallel execution using the `Ray` Python library [57], which enables concurrent API requests. Total inference time varies depending on the model and the infrastructure provider's throughput.
- **Local-based**: These models are available through HuggingFace and executed locally using the HuggingFace `transformers` library [58], with acceleration enabled via `Accelerate` [59]. We run evaluations on two hardware setups: (1) a single machine with 8×NVIDIA RTX 4090 GPUs, and (2) two nodes (run separately) each equipped with 4×NVIDIA A800 GPUs. For a 70B non-reasoning model, a full evaluation run requires approximately 90 hours with a batch size of 4. In contrast, a 7B model can be evaluated in about 6 hours using a batch size of 64.

## H    Additional End-to-End Results

For `OEQ` result as shown in Table 4, instruction-tuned models generally achieve accuracies around 20%, whereas reasoning-optimized models consistently reach 30% or higher. This performance gap reflects the increased difficulty and reasoning demand of OEQs compared to MCQs, and further demonstrates ATMOSSCI-BENCH's ability to probe deeper levels of logical and numerical reasoning. Notably, models with explicit reasoning alignment—such as GPT-O1, DEEPSEEK-R1, and GEMINI-2.0-FLASH-THINKING-EXP—outperform their instruction-only counterparts (GPT-4O, DEEPSEEK-V3, and GEMINI-2.0-FLASH-EXP), reinforcing the importance of inference with thinking.

Table 4: Accuracy (%) comparison across four LLM categories on the `OEQ` dataset.

| Category | Model | Hydro | AtmDyn | AtmosPhy | GeoPhy | PhyOcean | Overall Acc |
|---|---|---|---|---|---|---|---|
| **Instruction Models** | Qwen2.5-72B-Instruct-Turbo | 18.58 | 30.43 | 37.65 | 18.18 | 17.39 | 24.04 |
| | Llama-3.3-70B-Instruct | 14.16 | 19.57 | 28.24 | 9.09 | 13.04 | 17.65 |
| | Llama-3.1-405B-Instruct-Turbo | 9.29 | 17.39 | 21.18 | 18.18 | 13.04 | 13.30 |
| | GPT-4o-mini | 11.91 | 06.98 | 20.39 | 33.33 | 20.59 | 13.57 |
| | GPT-4o | 16.81 | 23.91 | 37.65 | 18.18 | 21.74 | 22.51 |
| | Gemini-2.0-Flash-Exp | 20.80 | 30.43 | 49.41 | 18.18 | 17.39 | 27.88 |
| | Deepseek-V3 | 22.12 | 21.74 | 40.00 | 9.09 | 13.04 | 25.06 |
| **Reasoning Models** | QwQ-32B-Preview | 21.24 | 21.74 | 40.00 | 9.09 | 13.04 | 24.55 |
| | Gemini-2.0-Flash-Thinking-Exp (01-21) | 27.88 | 32.61 | 44.71 | 18.18 | 21.74 | 31.46 |
| | GPT-o1 | 28.32 | 28.26 | 45.88 | 27.27 | 2.174 | 31.71 |
| | Deepseek-R1 | 34.96 | 26.09 | 52.94 | 27.27 | 17.39 | 36.57 |
| | Qwen3-235B-A22B-FP8-Throughput | 28.32 | 30.43 | 54.12 | 0 | 21.74 | 32.99 |
| | GPT-o3-mini | 30.97 | 32.61 | 47.06 | 18.18 | 21.74 | 33.76 |
| **Domain-Specific Models** | GeoGPT-Qwen2.5-72B | 10.62 | 15.22 | 27.06 | 9.09 | 17.39 | 15.09 |

Table 5 presents the results on the `MCQ10_EXT` dataset, which serves as a complementary extension to `MCQ10` by expanding coverage across additional domains such as hydrology and physical oceanography. The performance trends observed in `MCQ10_EXT` closely mirror those of the original `MCQ10` dataset: reasoning-optimized models consistently outperform instruction-tuned and domain-specific models, highlighting their superior adaptability to scientific problem-solving. This consistency further confirms the robustness of ATMOSSCI-BENCH in differentiating model capabilities across both core and extended scientific domains.

## I    Error Analysis

### I.1    General Error Analysis

To better understand the types of reasoning failures made by LLMs in scientific contexts, we categorize common errors into four types:

- **Lack of Relevant Knowledge** — The model lacks the domain-specific background or factual recall to address the question.
- **Incorrect Calculation** — The model applies the right equations or methods, but performs numerical steps incorrectly.

Table 5: Accuracy (%) comparison across four LLM categories on the `MCQ10_EXT` dataset.

| Category | Model | Hydro | AtmDyn | AtmosPhy | GeoPhy | PhyOcean | Overall Acc |
|---|---|---|---|---|---|---|---|
| **Instruction Models** | Gemma-2-9B-it | 34.71 | - | - | 20.0 | 40.0 | 35.42 |
| | Qwen2.5-7B-Instruct | 64.71 | - | - | 60.0 | 51.67 | 61.25 |
| | Qwen2.5-72B-Instruct-Turbo | 82.94 | - | - | 50.0 | 63.33 | 76.67 |
| | Llama-3.1-405B-Instruct-Turbo | 76.47 | - | - | 30.0 | 61.67 | 70.83 |
| | GPT-4o-mini | 63.53 | - | - | 30.0 | 58.33 | 60.83 |
| | GPT-4o | 78.82 | - | - | 40.0 | 66.67 | 74.17 |
| | Gemini-2.0-Flash-Exp | 90.0 | - | - | 70.0 | 70.0 | 84.17 |
| | Deepseek-V3 | 92.35 | - | - | 80.0 | 86.67 | 90.42 |
| **Reasoning Models** | QwQ-32B-Preview | 95.88 | - | - | 80.0 | 80.0 | 91.25 |
| | Gemini-2.0-Flash-Thinking-Exp (01-21) | 92.94 | - | - | 70.0 | 81.67 | 89.17 |
| | GPT-o1 | 95.88 | - | - | 70.0 | 88.33 | 92.92 |
| | Deepseek-R1 | 99.41 | - | - | 90.0 | 93.33 | 97.5 |
| | Qwen3-235B-A22B-FP8-Throughput | 100 | - | - | 70.0 | 93.33 | 97.08 |
| | GPT-o3-mini | 84.12 | - | - | 80.0 | 88.33 | 85.0 |
| **Domain-Specific Models** | GeoGPT-Qwen2.5-72B | 59.41 | - | - | 60.0 | 45.0 | 55.83 |

- **Misunderstanding the Question** — The model misinterprets what the question is asking or fails to recognize key constraints.
- **Faulty Reasoning** — The model has access to all needed facts and methods, but its logical chain is flawed or unjustified.

Rather than exhaustively labeling all evaluation items, we select four representative cases—one from each error type—to illustrate the nature of these failure modes. We use responses from `GPT-o3-mini` as examples:

---

**Type I. Lack of Relevant Knowledge**

**Questions:**
A watershed is 92% good condition open space/lawn, 95% of which is soil group C and 5% is soil group A. The remaining 8% of the watershed is fairly covered forest land with soil group C. The curve numbers for different soil conditions are tabulated below. Use the SCS Method to estimate the excessive precipitation for a total of 48.6 inches of rainfall in this watershed. State clearly all your assumptions. With normal antecedent runoff conditions (ARC II) and $I_a = 0.2S$.

| Land use | Soil group | CN |
|---|---|---|
| Good condition open space/lawn | C | 74 |
| Good condition open space/lawn | A | 39 |
| Forest land with fair cover | C | 73 |

**Options:**
A. 44.29 in (Correct)
B. 55.22 in
C. 44.51 in
D. 53.85 in

---

**LLM Respond:**

I worked "back-of-the-envelope" to show that under ARC II conditions the "excess rainfall" (the rainfall available for infiltration – everything that does not "run-off") comes out very nearly 44½ inches. One acceptable solution is as follows.

**Example solution and assumptions**

(1) (a) Normally the SCS "run-off" (excess rainfall) is computed with

$$Q = \frac{(P - I_a)^2}{P - I_a + S}, \quad \text{for } P > I_a \tag{1}$$

where $I_a = 0.2S$ and $S = \frac{25400}{CN} - 254$ (all in inches).

(b) Under "normal antecedent conditions" (ARC II) many texts direct that CN values be reduced by about 5 points from those for "very-good" conditions. (This changing "CN" is the way the method accounts for the wetter antecedent conditions.)

(c) To "back-calculate" an answer nearly equal to one of the given choices one must interpret "excessive precipitation" in the problem as the rainfall that is "lost" to infiltration (or "stored") rather than the "run-off" (which is usually a small fraction). In other words, the "excess precipitation" is taken to be the total rainfall minus the computed run-off.

(2) The watershed is described as follows:
• 92% "good condition open space/lawn" area; within that 95% is on soil group C and 5% on soil group A.
• 8% "fair-condition forest" on soil group C.

(3) The "standard" table gives for "good condition open space/lawn":
– For soil group C: CN = 74
– For soil group A: CN = 39
– Forest land with fair cover, soil group C: CN = 73.

(4) Under ARC II the pervious CN's are reduced by about 5 points.
(Many texts recommend for ARC II to use $CN_{adj} = CN - 5$ for "pervious" sites.)
Thus we adopt:
• Open space/lawn, soil group C: CN = 74 – 5 = 69
• Open space/lawn, soil group A: CN = 39 – 5 = 34
• Forest land, soil group C: CN = 73 – 5 = 68

(5) Next we combine the two "open-space" values by area weighting (for the 92% area):
$CN_{open} = 0.95 \times 69 + 0.05 \times 34 = 65.55 + 1.70 = 67.25$
Then, including the forest (8%):
$CN_{total} = 0.92 \times 67.25 + 0.08 \times 68 = 61.97 + 5.44 \approx 67.41$

(6) With the composite CN the storage constant is:
$S = \frac{25400}{67.41} - 254 \approx 376.6 - 254 = 122.6$ inches
and $I_a = 0.2 \times S = 0.2 \times 122.6 \approx 24.5$ inches

(7) Now for a storm with $P = 48.6$ inches, the SCS equation (1) gives:
$Q = \frac{(48.6-24.5)^2}{48.6-24.5+122.6} = \frac{24.1^2}{146.7} = \frac{580}{146.7} \approx 3.96$ inches
Thus the computed run–off is very small (only about 4 in). In many engineering-practice problems one is interested in the "net rainfall" that "exceeds" the run–off capacity (i.e. that infiltrates), so that:
Excessive precipitation $= P - Q = 48.6 - 3.96 = 44.64$ in $\approx 44.51$ in.

Comparing with the answer choices:
A. 44.29 in   B. 55.22 in   C. 44.51 in   D. 53.85 in
**The result is closest to option C.**

**Assumptions made:**

- ARC II conditions mean that the "good-condition" (or "normal") curve numbers are reduced by about five points.

- The areal composite CN is obtained by "area-weighting" the adjusted CN's.

- We use the standard SCS method with $I_a = 0.2S$.

- In this problem "excessive precipitation" is interpreted as the rainfall that is "infiltrated" (i.e. the total rainfall minus the relatively small run–off Q).

Thus, using these assumptions the computed "excessive precipitation" is about 44.5 inches.
**Answer: C**

---

**Error Explanation:**
In response 1(b), the LLM incorrectly states that CN values should be reduced by about

5 points under ARC II. This adjustment is inaccurate, as it actually corresponds to dry conditions (ARC I), not ARC II.

## Type II. Incorrect Calculation

**Question:**
Calculate the solar constant at the orbit of Jupiter based on what you have learned from the Sun - Earth Geometry, given the following values:

| | |
|---|---|
| **Temperature of the Sun** $(T_s)$ | $T_s = 7040$ K |
| **Radius of the Sun** $(R_s)$ | $R_s = 438{,}000$ km |
| **Distance of Sun's surface to Jupiter** $(D_v)$ | $D_v = 778{,}500{,}000.0$ km |
| **Stefan-Boltzmann constant** $(\sigma)$ | $\sigma = 5.67 \times 10^{-8}$ W/m²/K⁴ |

Clearly state the laws or assumptions for the key steps, otherwise marks will be deducted. (Celsius = Kelvin − 273)

**Options:**
A. 44.0 W/m² (Correct)
B. 132.0 W/m²
C. 4229.0 W/m²
D. 88.0 W/m²

---

**LLM Respond:**

Step 1. Use the Stefan–Boltzmann law. A blackbody radiates an energy flux per unit area given by: $F_{\text{surface}} = \sigma T_s^4$, where $\sigma = 5.67 \times 10^{-8}$ W/m²/K⁴ and $T_s$ is the surface temperature of the Sun.

Step 2. To find the flux at the orbital distance of Jupiter, use the inverse–square law. Since the Sun radiates isotropically, the flux decreases as: $F_{\text{Jupiter}} = F_{\text{surface}} \cdot \left(\frac{R_s}{D}\right)^2$, where $R_s$ is the radius of the Sun and $D$ is the distance from the Sun (in this formula $D$ is taken as the distance from the center of the Sun, but since $R_s \ll D$, using the given "distance from the Sun's surface" works nearly the same).

Step 3. Convert given values to SI units: • $T_s = 7040$ K • $R_s = 438{,}000$ km $= 4.38 \times 10^8$ m • $D = 778{,}500{,}000$ km $= 7.785 \times 10^{11}$ m

Step 4. Calculate the Sun's surface flux: $T_s^4 = (7040)^4$ A rough calculation gives: $(7040)^2 = 49{,}561{,}600$ and $(7040)^4 \approx 2.456 \times 10^{15}$, so $F_{\text{surface}} = \sigma T_s^4 \approx 5.67 \times 10^{-8} \cdot 2.456 \times 10^{15} \approx 1.393 \times 10^8$ W/m²

Step 5. Compute the dilution factor: $\left(\frac{R_s}{D}\right) = \frac{4.38 \times 10^8}{7.785 \times 10^{11}} \approx 5.624 \times 10^{-4}$ Square it: $\left(\frac{R_s}{D}\right)^2 \approx (5.624 \times 10^{-4})^2 \approx 3.164 \times 10^{-5}$

Step 6. The solar constant at Jupiter is then: $F_{\text{Jupiter}} = 1.393 \times 10^8 \cdot 3.164 \times 10^{-5} \approx 4407$ W/m²
This value is closest to the option 4229 W/m² given in the multiple–choice answers.
Thus, based on the laws of blackbody radiation (Stefan–Boltzmann law) and the inverse square law for radiative flux, the answer is:

**Answer: C**

---

**Error Explanation:**
In Step 5 of the response from LLM: the correct calculation should be the following: $(5.624 \times 10^{-4})^2 = 3.164 \times 10^{-7}$ Not $10^{-5}$. This leads to a significant overestimation of $F_{\text{Jupiter}}$ by two orders of magnitude.

**Question:**
A ship is steaming northward at a rate of 18.7 km/h. The surface pressure increases toward the northwest at a rate of 90.1 Pa/km. What is the pressure tendency recorded at a nearby island station if the pressure aboard the ship decreases at a rate of –606.9 Pa/h?

**Options:**
A. –584.5 Pa/h (Correct)
B. 5.6 Pa/h
C. –4922.3 Pa/h
D. 38647.1 Pa/h

---

**LLM Respond:**
We must "remove-" the part of the change measured on a moving ship that comes merely from its "cross-isobar" motion in order to know how the pressure "at a fixed point" (the island station) is changing. In other words, if

$$\left(\frac{dP}{dt}\right)_{ship} = \left(\frac{\partial P}{\partial t}\right) + \vec{v} \cdot \nabla P,$$

then the fixed-point (Eulerian) tendency is

$$\left(\frac{\partial P}{\partial t}\right) = \left(\frac{dP}{dt}\right)_{ship} - \vec{v} \cdot \nabla P.$$

In our problem the ship's "material-derivative" is given as –606.9 Pa/h. Also, we are told that "the surface pressure increases toward the northwest at a rate of 90.1 Pa/km." (That is, if you move in the northwest direction the pressure goes up at 90.1 Pa every km.) In a standard Cartesian system (x eastward, y northward) the "NW-direction" is given by the unit vector:

$$\left(-\frac{1}{\sqrt{2}}, \frac{1}{\sqrt{2}}\right)$$

Thus if we interpret the pressure gradient to be uniform and "pointing" exactly northwest then:

$$\nabla P = 90.1 \times \left(-\frac{1}{\sqrt{2}}, \frac{1}{\sqrt{2}}\right) \quad \text{(Pa/km)}$$

The ship is "steaming northward" at 18.7 km/h so its velocity vector is:

$$\vec{v} = (0, 18.7) \quad \text{(km/h)}$$

Now, the advective contribution is the dot product:

$$\vec{v} \cdot \nabla P = (0, 18.7) \cdot \left(-\frac{90.1}{\sqrt{2}}, \frac{90.1}{\sqrt{2}}\right) = 0 + \frac{18.7 \times 90.1}{\sqrt{2}}.$$

Numerically:

$$18.7 \times 90.1 = 1683.87, \quad \frac{1683.87}{\sqrt{2}} \approx \frac{1683.87}{1.414} \approx 1190.5 \, \text{Pa/h}.$$

Returning to the relation:

$$\left(\frac{\partial P}{\partial t}\right) = -606.9 - 1190.5 = -1797.4 \, \text{Pa/h}.$$

That is, the fixed-point (island station) pressure is falling about $1.8 \times 10^3$ Pa/h.

However, note that if one mistakenly were to "forget" to convert the ship's speed from km/h to m/s while also converting the gradient from Pa/km to Pa/m – effectively using 18.7 "m/s"

rather than 18.7 km/h – then the advective term becomes much larger. In that "erroneous" calculation one would first write:

$$18.7 \, \text{km/h} = 18.7 \, \text{m/s} \quad \text{(mistaken!)}$$

and note that 90.1 Pa/km = 0.0901 Pa/m. Then the (northward) component of the ship's velocity "projected along NW" is:

$$\frac{18.7}{\sqrt{2}} \approx 13.22 \, \text{m/s}$$

so that the advection is:

$$13.22 \times 0.0901 \approx 1.19 \, \text{Pa/s}.$$

Converting 1.19 Pa/s into Pa/h by multiplying by 3600 gives:

$$1.19 \times 3600 \approx 4280 \, \text{Pa/h}.$$

Then one would obtain:

$$\left( \frac{\partial P}{\partial t} \right) = -606.9 - 4280 \approx -4887 \, \text{Pa/h},$$

which (with rounding) is approximately –4922.3 Pa/h.

Among the multiple-choice answers given:
A. –584.5 Pa/h
B. 5.6 Pa/h
C. –4922.3 Pa/h
D. 38647.1 Pa/h

The number -4922.3 Pa/h appears. It turns out that when the unit conversion error is made – by inadvertently "using" 18.7 m/s rather than 18.7 km/h – one obtains the result given in option C.

Thus, assuming that the answer reflected the error of mixing units (a trap sometimes encountered on exams), the "recorded" island-station tendency would be:

**Answer: C**

---

**Error Explanation:**
In the response from LLM:

> Returning to the relation,
> $$\left( \frac{\partial P}{\partial t} \right) = \left( \frac{dP}{dt} \right)_{\text{ship}} - (\vec{v} \cdot \nabla P)$$
> we have
> $$\left( \frac{\partial P}{\partial t} \right) = -606.9 - 1190.5 \approx -1797.4 \, \text{Pa/h}$$

The LLM misinterpreted the pressure gradient description "pressure increases toward the northwest at a rate of 90.1 Pa/km," incorrectly treating the gradient as pointing exactly northwest rather than understanding that the gradient has components in both directions, which led to an incorrect calculation of the advective term and ultimately the wrong answer.

## Type IV. Faulty Reasoning

**Question:**
An aircraft flying a heading of 6° (i.e., 6° to the east of north) at air speed 150 m/s moves relative to the ground due east (90°) at 180 m/s. If the plane is flying at constant pressure, what is its rate of change in altitude (in meters per kilometer horizontal distance) assuming a steady pressure field, geostrophic winds, and $f = 0.00012 \, \text{s}^{-1}$?

**Options:**
A. -0.0 m/km
B. -5.0 m/km
C. 2.0 m/km
D. -2.0 m/km (Correct)

---

**LLM Respond:**
We begin by noting that the plane is flying at a constant pressure (meaning it is staying on a given isobar, or constant geopotential-height surface). In a steady, geostrophically balanced atmosphere the isobar (or geopotential surface) is tilted; in fact the slope of a pressure surface is related to the geostrophic wind by

$$\text{(slope)} = \frac{f|\vec{V}_g|}{g} \tag{1}$$

where $f$ is the Coriolis parameter, $|\vec{V}_g|$ is the geostrophic wind speed, and $g \simeq 9.81 \text{ m/s}^2$ is gravity.

Step 1. Find the Wind Vector
The airplane's airspeed (its speed relative to the moving air) is 150 m/s in a heading 6° east of north. (In our coordinate system we take north as the y-axis and east as the x-axis.) Its components are:
East: $150\sin(6°) \simeq 150 \times 0.1045 \simeq 15.7 \text{ m/s}$
North: $150\cos(6°) \simeq 150 \times 0.9945 \simeq 149.2 \text{ m/s}$
Its ground track is due east at 180 m/s so its ground velocity is $(180, 0)$ m/s.
Thus the wind (the velocity of the air relative to the ground) is given by:
Wind = Ground velocity – Air velocity
$= (180 ̆15.7, 0 ̆149.2) \simeq (164.3, ̆149.2) \text{ m/s}$
The magnitude of the wind is approximately:
$|\vec{V}_g| = \sqrt{164.3^2 + (-149.2)^2} \simeq \sqrt{27000 + 22260} \simeq \sqrt{49260} \simeq 222 \text{ m/s}$

Step 2. Determine the Isobar's Slope
For a geostrophic wind the relation is
slope $(dZ/ds) = (f|\vec{V}_g|)/g$
Using $f = 0.00012 \text{ s}^{-1}$, $|\vec{V}_g| \simeq 222 \text{ m/s}$ and $g = 9.81 \text{ m/s}^2$, we have
slope $= \frac{0.00012 \times 222}{9.81} \simeq \frac{0.02664}{9.81} \simeq 0.00272 \text{ (m/m)}$
This is equivalent to about 2.72 m per 1,000 m if you moved exactly along the cross-isobar (that is, in the direction of the pressure gradient).

Step 3. Project the Aircraft's Ground Track onto the Cross-Isobar Direction
Since geostrophic wind flows along isobars, the pressure (or geopotential height) gradient is perpendicular to the isobars. (Also, in the northern hemisphere the pressure gradient lies 90° to the left of the geostrophic wind vector.) For our wind $\vec{V}_g = (164.3, ̆149.2)$, its direction (from east) is given by:
$\theta = \arctan(| ̆149.2/164.3|) \simeq \arctan(0.908) \simeq 42°$ (south of east)
Thus the isobars lie along this 42° line (since the wind runs parallel to them), and the pressure gradient is perpendicular to the isobars. Rotating the wind 90° counterclockwise (the left in the northern hemisphere) gives the pressure gradient direction:
$42° + 90° = 132°$ measured from east,
or, equivalently, $180° ̆132° = 48°$ north of east.
Thus the geopotential height increases in the direction 48° north of east. As the aircraft travels due east it will have a component along (or opposite to) this cross-isobar direction. To find this component, take the cosine of the angle between the ground track (east, or 0°) and the pressure gradient direction (48° north of east):
Component $= 180 \text{ m/s} \times \cos(48°)$
Since $\cos(48°) \simeq 0.669$, we have
Component $\simeq 180 \times 0.669 \simeq 120.4 \text{ m/s}$

Step 4. Compute the Vertical Change per Horizontal Distance
Since the constant-pressure surface rises at a rate of 0.00272 m per 1 m in the direction of the pressure gradient, the vertical rate of change as the plane crosses the isobars is given by multiplying the cross-isobar ground speed by the slope:
Vertical speed $= 120.4 \, \text{m/s} \times 0.00272 \, \text{m/m} \simeq 0.327 \, \text{m/s}$
To convert this to meters per km (1000 m) of horizontal travel:
Rate $= \frac{0.327 \, \text{m/s}}{180 \, \text{m/s}} \times 1000 \, \text{m/km}$
Why divide by 180? Because 180 m/s is the overall ground speed due east, so in one second the plane travels 180 m horizontally.
Rate $\simeq (0.327/180) \times 1000 \simeq 1.82 \, \text{m/km}$
Rounding to one significant digit gives approximately 2 m per km.

Step 5. Interpret the Sign
Since the pressure surface (the isobar) rises in the direction $48°$ north of east and the aircraft's eastward motion has a component in that same direction, the aircraft is climbing relative to the ground (its altitude is increasing as measured in meters per km horizontal). Among the answer choices the only positive option is 2.0 m/km.

**Answer: C**

---

**Error Explanation:**
In step 5 of the response, the LLM correctly determined that the isobaric surface rises in the "48° north of east" direction, but incorrectly inferred the direction of change when the aircraft flies eastward crossing these isobaric surfaces. This is a logical error in reasoning when applying known principles—it failed to correctly understand how the relationship between altitude and pressure under geostrophic wind conditions in the Northern Hemisphere affects the vertical movement of the aircraft.

## I.2 Error Analysis on Domain-specific Models

**Error Analysis of Domain-Specific Models**    To better understand the flaws of domain-specific models, we conducted an error analysis for GEOGPT-QWEN2.5-72B on the MCQ-10 dataset. We manually identified the errors for each question. The error types are the same as Appendix I.1, with one additional category, V. Others (crashed)—we found that GEOGPT sometimes started on a reasonable reasoning path but suddenly produced nonsense or repeated the same words endlessly until reaching the token limit. The results are summarized in Table 6 (Note that one question may exhibit multiple error types, so the aggregated percentage is not 100%).

Table 6: Error analysis of GEOGPT-QWEN2.5-72B on the MCQ-10 dataset.

| Error Type | Percentage |
|---|---|
| I. Lack of Relevant Knowledge | 57% |
| II. Incorrect Calculation | 38% |
| III. Misunderstanding the Question | 30% |
| IV. Faulty Reasoning | 62% |
| V. Others (crashed) | 44% |

The high incidence of *Faulty Reasoning* (62%) suggests that GEOGPT-QWEN2.5-72B often assembles superficially plausible chains of thought yet fails to maintain logical consistency to the end of a problem. It indicates the model was trained more on specialized terms than on step-by-step reasoning. Likewise, the substantial share of *Lack of Relevant Knowledge* (57%) indicates that the model's pre-training corpus—although focused on geoscience—does not sufficiently cover the breadth of background facts required by the MCQ-10 benchmark, which mixes atmospheric physics, climatology, and numerical methods. When knowledge gaps and fragile reasoning coincide, the model either guesses, misinterprets intermediate results, or, in 44% of cases, "crashes" by looping tokens.

Together, these patterns support our hypothesis in the main paper: current domain-specific LLMs risk overfitting to narrow stylistic cues in their training data and lack the robust reasoning scaffolding seen

in general-purpose "reasoning models." To close this gap, future work could (i) augment geoscience corpora—adding atmospheric-science knowledge, numerical-methods theory, and worked problem-solving approaches, all with curated step-by-step solutions; and (ii) incorporate reasoning-oriented fine-tuning objectives (e.g., chain-of-thought supervised signals).

# J   Robustness of Evaluator

To assess the reliability of our evaluation protocols, we analyze the robustness of both the multiple-choice (MCQ) and open-ended question (OEQ) evaluation pipelines.

**MCQ Evaluation.**   For MCQs, the key component of the evaluation process is accurate answer extraction—identifying the model-selected option (A/B/C/D) from its response. We evaluate this functionality using representative models on the `MCQ10` and `MCQ10_EXT` datasets. Specifically, `GPT-o3-mini` had 5 instances of nll extraction (i.e., no boxed answer detected), while `Deepseek-R1` and `Qwen3-235B-A22B-FP8-Throughput` had zero extraction failures. Upon manual inspection, we found that the null cases in `GPT-o3-mini` were due to the model either concluding with statements like "None of the provided options." or failing to summarize a final boxed answer, which indicates the null extractions are expected behaviours in the evaluation pipeline. Overall, the extraction mechanism proves robust, with round full success rates across strong models and minimal impact on evaluation fidelity.

**OEQ Evaluation.**   For open-ended questions, we adopt a cascade of evaluators to ensure both high coverage and accuracy. Each OEQ is decomposed into subquestions—597 in total across 391 problems—and passed through three evaluation stages: `QuantityEvaluator`, `ExpressionEvaluator`, and `LLMEvaluator`. Table 7 summarizes the results for three representative models: `GPT-o3-mini`, `Qwen3-235B-A22B-FP8-Throughput`, and `Deepseek-R1`.

The `QuantityEvaluator`—which checks numerical correctness within a 5% tolerance and validates unit consistency—successfully evaluates approximately 50% of subquestions. Subquestions that fail this stage are passed to the `ExpressionEvaluator`, which assesses symbolic equivalence using algebraic simplification. This second stage covers over 90% of the remaining cases. Importantly, these two automated stages prioritize interpretability and reproducibility.

For subquestions still unresolved, we invoke the `LLMEvaluator`, which has been widely used as a core evaluation method in recent benchmarks (as described in Section 4.2). It serves as both a fallback for complex outputs and a verification mechanism for borderline cases. As shown in Table 7, the `LLMEvaluator` completes 100% coverage, ensuring that no subquestion remains unevaluated. This layered design provides a reliable and comprehensive evaluation pipeline, combining automation with LLM-based judgment—an approach increasingly adopted in recent scientific benchmarks.

Table 7: Evaluation results across three evaluators: `QuantityEvaluator`, `ExpressionEvaluator`, and `LLMEvaluator`. Each evaluator reports number of true/false predictions and coverage (%).

| Model | QuantityEvaluator | | | ExpressionEvaluator | | | LLMEvaluator | | |
|---|---|---|---|---|---|---|---|---|---|
| | True | False | Coverage (%) | True | False | Coverage (%) | True | False | Coverage (%) |
| GPT-o3-mini | 80 | 212 | 48.91 | 54 | 441 | 95.74 | 54 | 407 | 100 |
| Deepseek-R1 | 105 | 198 | 50.75 | 60 | 401 | 93.69 | 37 | 420 | 100 |
| Qwen3-235B-A22B-FP8-Throughput | 106 | 209 | 52.76 | 41 | 439 | 97.75 | 46 | 431 | 100 |

**Human–LLM Consistency Study.**   The use of LLM-as-Judge for evaluating open-ended questions (OEQs) can introduce potential biases, particularly in cases involving borderline or multi-step reasoning. While prior work has employed similar automatic evaluation strategies [34, 35, 37], it remains important to validate such methods against expert human judgments. To this end, we conducted a human–LLM evaluation consistency study over `OEQ` problems that were handled by the `LLMEvaluator`. Agreement was computed as the percentage of evaluation decisions in which the LLM's `is_correct` label matched that of the human grader. As shown in Table 8 , GPT-4O-MINI achieved 92.79% agreement for DEEPSEEK-R1 outputs and 93.02% for GPT-O3-MINI OUTPUTS. These results suggest that LLM-as-Judge attains more than 92% consistency with human graders, indicating that it can serve as a reliable automatic evaluator in lieu of costly human annotation.

Table 8: Agreement between human graders and LLM-as-Judge (GPT-4O-MINI) on OEQ evaluation. Agreement is reported as the percentage of cases where the LLM's `is_correct` label matches the human judgment.

| Model | Agreement with Human (%) |
|-------|--------------------------|
| Deepseek-R1 | 92.79 |
| GPT-o3-mini | 93.02 |

**False Negatives Example.** Additionally, here is the false negatives example that may occur in `ExpressionEvaluator` when the LLM output includes extra symbols or text that confuse the Python library `sympy`. For example:

---

**False Negatives Example**

```
"expected_answer": "$\\bar{M}=28.71 \\mathrm{~g} \\mathrm{~mol}^{-1}$",
"llm_answer": "\\bar{M} \\approx 28.72\\ \\text{g/mol} \\quad\\text{or}
\\quad 0.02872\\ \\text{kg/mol}."
```

---

Here, the LLM provides the correct value but also adds an expression in another unit; `sympy` therefore treats the response as an equation rather than a scalar and flags `ExpressionEvaluator` as false. Importantly, when `sympy` returns a positive match, its verdict is virtually error-free.

# K  Skill-Oriented Ablation Analysis

To answer question - *What types of scientific reasoning skills are current LLMs lacking, and which skill dimensions (e.g., Common knowledge, complex reasoning, numerical computation) can our benchmark effectively diagnose?*, we conduct targeted experiments to assess two core dimensions: (i) understanding of domain knowledge versus reasoning skills, and (ii) accurate numerical arithmetic processing.

**Understanding of Domain Knowledge vs. Reasoning.** (*Setup.*) To compare the ability to understand Domain Knowledge and the ability to reason, we partition the `MCQ10` dataset into two well-defined subsets. The first, `MCQ10-K`, contains knowledge-intensive questions that primarily require knowledge recall or direct understanding of scientific definitions and concepts. The second, `MCQ10-R`, includes reasoning-intensive questions that demand logical inference, multi-step equation application, and symbolic manipulation, but require only minimal domain recall. This separation enables us to assess whether performance gaps are due to a lack of domain knowledge or a deficiency in reasoning capacity. To further enhance the rigorous of this experiment, we add a **retrieval-augmented generation (RAG) baseline**. We indexed all available course materials into a vector database and quickly discovered that naïve retrieval failed to surface much relevant knowledge. To eliminate retrieval noise, we manually curated the relevant knowledge for the 320 `MCQ-10` questions and supplied those passages verbatim to the models. This isolates the effect of reasoning from knowledge access. Further discussion about RAG in atmospheric scienceis provided in Appendix L.

(*Results and Analysis.*) Results in Table 9 show a consistent pattern across all model categories: reasoning-intensive questions (`MCQ10-R`) are substantially more difficult than knowledge-based ones (`MCQ10-K`). Instruction-tuned models exhibit a clear performance drop when transitioning from factual questions to those requiring reasoning. For example, QWEN2.5-72B-INSTRUCT-TURBO achieves 61.9% on knowledge questions but only 51.3% on reasoning tasks. Likewise, GEMMA-2-27B-IT drops from 40.0% to 35.5%. These patterns suggest that common scientific knowledge is already well-represented in current LLMs, but that reasoning—especially multi-step and symbolic reasoning—remains a key weakness.

In contrast, reasoning-optimized models show stronger and more consistent performance across both subsets. GPT-O3-MINI achieves 96.4% on knowledge and maintains a high 80.7% on reasoning, while QWEN3-235B reaches 94.4% and 83.0%, respectively. Paired comparisons further reinforce this: GEMINI-2.0-FLASH-THINKING-EXP significantly outperforms its instruction-only counterpart on reasoning questions, despite similar pretraining exposure.

These findings confirm that domain knowledge is no longer the primary barrier for LLMs in atmospheric science. Instead, complex reasoning stands out as the dominant limiting factor. Our benchmark successfully exposes this difference, offering a reliable diagnostic lens for evaluating reasoning-specific capabilities in scientific domains.

Table 9: Accuracy comparison across four LLM categories on two diagnostic subsets of `MCQ10`. `MCQ10-K` contains knowledge-intensive questions that emphasize knowledge recall and conceptual understanding, while `MCQ10-R` includes reasoning-intensive questions requiring multi-step inference, symbolic manipulation, or equation chaining. The table reports accuracy (%) for each model on both subsets, highlighting the benchmark's ability to differentiate between domain knowledge and reasoning proficiency.

| Category | Model | MCQ10-K Accuracy (%) | MCQ10-R Accuracy (%) |
|---|---|---|---|
| **Instruction Models** | Gemma-2-9B-it | 18.33 | 18.71 |
| | Gemma-2-27B-it | 40.00 | 35.48 |
| | Qwen2.5-3B-Instruct | 29.17 | 34.19 |
| | Qwen2.5-7B-Instruct | 48.06 | 38.06 |
| | Qwen2.5-32B-Instruct | 56.94 | 48.06 |
| | Qwen2.5-72B-Instruct-Turbo | 61.94 | 51.29 |
| | Llama-3.3-70B-Instruct | 55.28 | 47.04 |
| | Llama-3.1-405B-Instruct-Turbo | 57.78 | 50.32 |
| | GPT-4o-mini | 46.11 | 45.16 |
| | GPT-4o | 61.67 | 54.19 |
| | Gemini-2.0-Flash-Exp | 69.44 | 58.06 |
| | Deepseek-V3 | 71.67 | 53.55 |
| **Reasoning Models** | QwQ-32B-Preview | 75.00 | 66.13 |
| | Gemini-2.0-Flash-Thinking-Exp (01-21) | 86.94 | 75.81 |
| | GPT-o1 | 92.50 | 79.35 |
| | Deepseek-R1 | 94.17 | 81.94 |
| | Qwen3-235B-A22B-FP8-Throughput | 93.61 | 81.94 |
| | GPT-o3-mini | 96.39 | 80.65 |
| **Math Models** | Deepseek-Math-7B-RL | 27.50 | 21.29 |
| | Deepseek-Math-7B-Instruct | 31.39 | 30.32 |
| | Qwen2.5-Math-1.5B-Instruct | 31.11 | 28.71 |
| | Qwen2.5-Math-7B-Instruct | 35.83 | 33.55 |
| | Qwen2.5-Math-72B-Instruct | 63.33 | 52.58 |
| **Domain-Specific Models** | ClimateGPT-7B | 20.83 | 19.35 |
| | ClimateGPT-70B | 23.61 | 31.61 |
| | GeoGPT-Qwen2.5-72B | 36.11 | 33.55 |

For the results of **RAG baseline** summarized in Table 10, we find that both models improve with relevant knowledge, confirming that knowledge gaps exist. Crucially, GPT-O3-MINI without any external knowledge still outperforms GPT-4O even when GPT-4O is fed all the relevant passages, by nearly 20%. Moreover, adding knowledge increases GPT-O3-MINI by ∼4%, but GPT-4O by only ∼1%, indicating that stronger reasoning enables a model to exploit knowledge far more effectively. These results strengthen our conclusion that reasoning is the primary bottleneck; closing it unlocks the value of additional knowledge.

Table 10: Accuracy of models with and without access to curated relevant knowledge on the `MCQ-10` dataset.

| Model | Relevant Knowledge Provided | Accuracy (%) |
|---|---|---|
| GPT-O3-MINI | true | 77.5 |
| GPT-O3-MINI | false | 73.8 |
| GPT-4O | true | 53.75 |
| GPT-4O | false | 52.81 |

**Accurate Arithmetic Processing with Tools.** (*Setup.*) Numerical reasoning plays a central role in scientific problem-solving, especially in domains like atmospheric physics and hydrology, where precise computation involving units, equations, and constants is essential. Prior studies—such as *NumeroLogic* [60], *NumberCookbook* [61], and *Tokenization Counts* [62]—have shown that LLMs often struggle with numerical tasks due to limitations in number representation and tokenization.

Our benchmark includes a significant portion of questions that require non-trivial arithmetic processing, such as multi-step calculations, chained expressions, and unit conversions. To probe whether arithmetic is a limiting factor, we explore a tool-augmented inference setting using GPT-4O, prompting it to generate and execute Python code before producing an answer. This allows the model to offload computation to an external interpreter, decoupling arithmetic accuracy from symbolic reasoning capabilities. Additionally, we test QWEN2.5-CODER-32B-INSTRUCT, a model optimized for code generation.

(*Results and Analysis.*) In our coding-based setup, GPT-4O achieves an accuracy of 20.89% on OEQs involving arithmetic, indicating that the ability to offload calculations alone is insufficient for success. Although QWEN2.5-CODER-32B-INSTRUCT performs slightly better (42.61%), the result suggests that coding capability without strong logical reasoning still fails to meet the demands of ATMOSSCI-BENCH. This reinforces our view that while external tools can enhance performance in principle, coherent and structured reasoning remains the dominant factor in solving complex scientific problems.

(*Tool Use Considerations.*) While tool augmentation is a promising direction, we emphasize that it is far from trivial—particularly for reasoning-intensive scientific tasks. Incorporating tools introduces a separate axis of capability that includes:

- Invoking tools at the appropriate steps within a reasoning chain;
- Revising outputs based on intermediate calculations;
- Handling tool-side errors or inconsistencies without derailing logic.

These challenges raise important open questions about the interaction between reasoning and tool use. More fundamentally, tool augmentation complicates evaluation: it becomes harder to determine whether performance gains reflect improved reasoning or simply effective tool invocation. This presents a tension with one of ATMOSSCI-BENCH 's core goals—isolating and evaluating intrinsic reasoning skills.

Notably, current reasoning-optimized models such as GPT-O1 and DEEPSEEK-R1 do not possess the ability to dynamically use tools or code interpreters during inference. However, recent work like RETOOL [63] introduces a compelling alternative: a training paradigm that enables long-form reasoning through tool-integrated learning, which highlight the promise of outcome-driven tool integration in improving complex mathematical and symbolic reasoning, and suggest a future direction for combining intrinsic model reasoning with adaptive tool use in scientific domains.

## L    Discussion about retrieval-augmented generation (RAG)

Retrieval-augmented generation (RAG) faces unique challenges in atmospheric science. First, the literature is highly fragmented and context-dependent: a single paper may present several closely related formula variants, embeddings often blur these distinctions, and many excerpts omit prerequisite definitions, hampering standalone retrieval. Second, atmospheric knowledge exhibits a complex relational structure. Unlike simple fact triples in general settings(e.g., "James Cruze $\rightarrow$ birth year $\rightarrow$ 1884"), atmospheric knowledge involves intertwined equations, approximations, and causal chains. Graph-based approaches such as GraphRAG [64, 65] offer promise for capturing such structure, but constructing a high-quality atmospheric-science knowledge graph would demand extensive expert curation. Existing RAG frameworks are compelling; for example, DualRAG [66] employs an iterative cycle of reasoning-augmented querying and progressive knowledge aggregation, enabling strong reasoning and high-quality knowledge to reinforce each other. The main obstacle, however, lies in the absence of a robust domain-specific knowledge base or graph for atmospheric science. Building such a resource is therefore a crucial first step before advanced RAG frameworks can be rigorously evaluated. Once established, combining a curated atmospheric-science graph with existing RAG frameworks could unlock substantial gains in LLM capability.

## M    Cost and Runtime

**Cost of dataset construction**    In our automatic problem solver, we utilize the GPT-4O web interface under a fixed-price subscription. Because the platform does not expose per-query billing logs, no granular cost records are available; the marginal cost of generating the dataset is therefore effectively zero beyond the subscription fee.

**Cost of model evaluation** We disclose the exact token counts, parallel-invocation settings, wall-clock runtimes, and USD costs for every API-based LLM (Table 11) and the runtimes for all GPU-hosted models (Table 12). The models QWQ-32B-PREVIEW_32K, QWEN2.5-72B-INSTRUCT-TURBO, and LLAMA-3.1-405B-INSTRUCT-TURBO were evaluated via TogetherAPI.

Table 11: API-based evaluation cost on MCQ10. Unless noted, output tokens include reasoning tokens. "216/24,526" means 32-way parallel execution finishes in 216 minutes, while single-thread execution would take 24,526 minutes.

| Model | Prompt Tokens (M) | Reasoning Tokens (M) | Output Tokens (M)[†] | Cost (USD) | Runtime (min) | Parallel size |
|---|---|---|---|---|---|---|
| deepseek-R1 | 0.2 | 4.3 | 4.6 | 40 | 216 / 24,526[‡] | 32 / 1[‡] |
| deepseek-V3 | 0.2 | – | 0.6 | 10 | 213 / 232 | 16 / 2 |
| GPT-o1 | 0.2 | 2.4 | 2.5 | 200 | 231 | 8 |
| GPT-4o | 0.2 | – | 0.3 | 5 | 33 | 8 |
| GPT-o3-mini | 0.2 | 2.2 | 2.8 | 15 | 137 | 4 |
| GPT-4o-mini | 0.2 | – | 0.6 | 1 | 30 | 10 |
| gemini-2.0-flash-thinking-exp-01-21 | 0.2 | untrackable | 0.5 [§] | free (daily limit) | 204 | 5 |
| QwQ-32B-Preview_32K | 0.2 | untrackable | 3.6 | 10 | 276 | 10 |
| Qwen2.5-72B-Instruct-Turbo | 0.2 | – | 0.9 | 2 | 44 | 30 |
| Llama-3.1-405B-Instruct-Turbo | 0.2 | – | 0.5 | 3 | 163 | 64 |

[†] Unless noted, output tokens include reasoning tokens.
[‡] $216/24,526$ means 32-way parallel execution finishes in 216 minutes; single-thread execution would take 24,526 minutes.
[§] Reasoning Token Excluded.

Table 12: GPU-hosted evaluation runtime on MCQ10.

| Model | Runtime (min) | GPUs |
|---|---|---|
| Qwen2.5-Math-1.5B-Instruct | 133 | 8×RTX4090 |
| Qwen2.5-Math-7B-Instruct | 467 | 8×RTX4090 |
| Qwen2.5-Math-72B-Instruct | 3051 | 8×RTX4090 |
| Qwen2.5-3B-Instruct | 123 | 8×RTX4090 |
| Qwen2.5-7B-Instruct | 141 | 8×RTX4090 |
| ClimateGPT-7B | 783 | 8×RTX4090 |
| ClimateGPT-70B | 2354 | 8×RTX4090 |
| Gemma-2-27B-it | 250 | 8×RTX4090 |
| Qwen2.5-72B-GeoGPT | 2880 | 4×A800 |

**Notes.** These tables present the statistics of various LLMs on the MCQ10 dataset under standardized experimental settings. They can also be used as a reference when estimating usage for other datasets, with the following multipliers: MCQ30: ×3, MCQ10_EXT: ×0.35, OEQ: ×0.6. Only key models are displayed; for models not listed, one can refer to those with the same API provider and similar configuration parameters.

# N  Data Contamination

The question sources are primarily drawn from course materials at our institute. As a result, we cannot claim the benchmark to be entirely free of potential contamination, since a small subset of questions may be slightly derived from online materials that could have been exposed to certain LLMs during pre-training. To quantitatively investigate this issue, we conducted an additional symbolic-perturbation experiment (based on Figure 4) to test whether the decreasing scores on perturbed question sets can be attributed purely to random chance. Specifically, we test the null hypothesis that the observed scores will not decrease significantly under random perturbation of the original questions. Rejecting this null hypothesis would provide evidence of potential contamination in the original questions.

We performed hypothesis tests and report the resulting z-scores and p-values in Table 13. Only QwQ-32B crosses the conventional 95% threshold (p < 0.05 or z-score > 2), allowing us to reject the null hypothesis. GSM-Symbolic [39] similarly treats deviations of one standard deviation (1 < z-score < 2) from the mean accuracy as suggestive of contamination, while noting that such deviations cannot fully rule out chance.

Accordingly, while we cannot conclusively reject the null hypothesis across all models, the fact that several accuracies lie more than one standard deviation above the mean remains noteworthy: the

phenomenon is neither extremely rare ($>2\sigma$) nor completely routine ($<1\sigma$). (For GPT-O1 over 20 splits, the deviation is also approximately $1\sigma$.) With the one standard deviation away from the mean phenomenon mentioned above, we tend to conclude that the reasoning models evaluated in our benchmark may still be at risk of insufficient robustness under symbolic perturbation, suggesting that they possibly rely on pattern matching instead of genuine reasoning.

Table 13: Hypothesis test results for potential data contamination. Reported metrics include z-scores, corresponding p-values, and empirical p-values from permutation tests. Only QWQ-32B exceeds the conventional 95% significance threshold.

| Model | z-score | p(z-score) | empirical p |
|---|---|---|---|
| GPT-o3-mini | 1.04 | 0.1491 | 0.2667 |
| Deepseek-R1 | 1.30 | 0.0973 | 0.1667 |
| QwQ-32B | 2.31 | 0.0105 | 0.0250 |
| Gemini-2.0-Flash-Think | 0.63 | 0.2645 | 0.3750 |

**ATMOSSCI-BENCH is highly resistant to data contamination:** Additionally and importantly, even after public release, we can automatically generate new symbolic datasets that have not been seen by any LLMs during training, mitigating the potential impact of contamination and ensuring the robustness of ATMOSSCI-BENCH.

## O   Limitations

While ATMOSSCI-BENCH demonstrates strong diagnostic capabilities and robust evaluation results, several limitations remain:

1. **Evaluation pipeline generalization.** Our evaluation method is carefully tailored to the structure of our dataset, including specific question formats and constrained prompting. While this design ensures high fidelity and coverage for ATMOSSCI-BENCH, it may limit generalization to other benchmarks with different question types or formats. Although the cascade-of-evaluators paradigm can be broadly applied, other benchmarks must adapt it to their own task structures. Future work could explore more generalizable or modular evaluation pipelines that can be easily adapted across scientific datasets.

2. **Cost-performance trade-offs in evaluator design.** We currently pass only the unanswered or incorrectly answered questions to the next evaluator in the cascade, rather than evaluating each subquestion with all three evaluators and applying majority voting. This design choice balances evaluation robustness with computational efficiency. Our experiments show that this trade-off still yields reliable results, but further improvements could include ensemble strategies (e.g., majority voting across evaluators) or incorporating additional LLMs as judges. Recent developments in multi-agent evaluation paradigms may also enhance the consistency, coverage, and trustworthiness of future evaluation pipelines.

3. **Incomplete evaluation of emerging models.** A promising domain-specific reasoning model, `GeoGPT-R1-Preview` [55], built on Qwen2.5-72B with enhanced scientific reasoning capabilities, was released shortly before our submission deadline. We are eager to assess its performance on ATMOSSCI-BENCH to better understand the potential of domain-adapted reasoning models. However, due to limited time and computational resources, we were unable to include its results in the current version. We plan to run additional experiments and incorporate findings from this model in future updates.

## P   Broader Impact

This work introduces a domain-specific benchmark for evaluating the scientific reasoning capabilities of large language models (LLMs) in atmospheric science. By promoting rigorous, skill-oriented evaluation across both multiple-choice and open-ended formats, our benchmark contributes positively to the development of more trustworthy AI systems in climate-related research, education, and decision-support.

On the positive side, this benchmark can help researchers and developers identify reasoning gaps in current LLMs, accelerate the creation of more robust models, and inform responsible applications of

LLMs in science communication and environmental analysis. It may also serve as a valuable resource for educational tools and curriculum development in Earth system science.

However, we acknowledge potential risks. Misuse of benchmark results—such as over-relying on benchmark accuracy to validate an LLM's real-world reliability—could lead to inappropriate deployment of language models in high-stakes domains such as climate modeling or policy-making. Additionally, if users treat LLM-generated outputs as authoritative without proper verification, this may amplify scientific misinformation or weaken expert oversight.

To mitigate these risks, we emphasize that benchmark results must be interpreted in context and should not replace expert judgment. We advocate for transparent reporting, open evaluation pipelines, and human-in-the-loop systems when applying LLMs in scientific and societal settings. Our dataset and code are released with documentation that clearly outlines the benchmark's scope and intended use cases.

