# OpenReview forum: "AtmosSci-Bench: Evaluating the Recent Advance of Large Language Model for Atmospheric Science"
_NeurIPS.cc/2025/Datasets_and_Benchmarks_Track — NeurIPS 2025 Datasets and Benchmarks Track poster_

### Official Review · Reviewer_WSaL · 2025-06-11

**Rating:** 5
**Confidence:** 3

**Summary:**

This paper introduces a benchmark dataset used to evaluate the performance of LLMs on questions relevant to atmospheric science (and closely related fields). The paper introduces this dataset, then measures the performance of various LLMs on the dataset.

The questions are drawn from atmospheric science courses at the authors' university. Unlike reference [33], which evaluated LLM knowledge and reasoning in atmospheric science by asking mostly conceptual questions (i.e., "What is the purpose of measurements at monitoring sites in relation to air pollutants and acid rain?"), this benchmark dataset tests mostly mathematical and symbolic reasoning abilities (i.e., "At $20^{\\circ} \\mathrm{C}$, glycerin occupies the area between two horizontal, parallel plates, each with an area of $9 \\mathrm{cm}^{2}$. The upper plate moves at a steady speed while being 1 cm above the stationary lower plate. The velocity gradient measured between these plates is $120 \\mathrm{~s}^{-1}$. Determine the shear force acting on the bottom plate.")

The dataset consists of multiple-choice questions as well as open-ended response questions. The authors include 67 unique multiple-choice questions, though they vary the numerical parameters 10 times for each question (so-called "symbolic perturbations"), so this creates a total of 670 multiple-choice questions in a dataset called "MCQ10". They also include an additional 240 multiple-choice questions in a different dataset they call "MCQ10_EXT". There are also a total of 391 unique open-ended questions, which do not involve symbolic perturbations, in a dataset called "OEQ".

I am not an atmospheric scientist, but I am a scientist in related domains. In my judgement, having looked at a sample of the questions by downloading the CSV files, the questions seem challenging and diverse.

While the reasoning models do well on the multiple-choice questions (the best models score close to 90%), the open-ended questions seem to be more difficult for the reasoning models (scoring no higher than 37%).

**Additional Feedback:**

I have not found any major flaws with this paper. I do have some minor concerns, but I think these can be resolved in the response period.

My most significant concern is minor concern #2. In my opinion, it suggests a lack of understanding of the purpose of symbolic variation.

In my judgement, this paper is a solid benchmark that deserves acceptance. It's very unclear to me how much it will benefit the atmospheric science community (besides testing the ability of students to use LLMs to cheat on their homework). However, it almost certainly has value as a challenging benchmark to test the symbolic and mathematical abilities of LLMs. This is especially true of the open-ended questions.

I have little doubt that LLMs will eventually saturate this benchmark once they add it to their training data (i.e., data leakage), but that doesn't mean it isn't a valuable benchmark nevertheless.

My recommendation is to accept the paper, though I am not extremely confident in that evaluation. Benchmarking LLMs is not my area of research, nor is atmospheric science.

**Dataset Code Accessibility:**

Yes

**Dataset Code Comments:**

There was some friction and difficulties in downloading and figuring out the dataset, but it was overall pretty straightforward.

I did find that following the instructions was not sufficient to download and run the dataset, and required some amount of background knowledge on my part.

For example, the first installation instruction was ```git clone https://github.com/your-username/atmossci-bench.git``` did not work, and replacing ```your-username``` with my github username did not work. Obviously downloading the repository from the anonymous github did work.

As another example, the second installation instruction was ```./setup.sh```, but I didn't have permission to run this script. I had to give myself permission with ```chmod +x setup.sh```.

The rest of the instructions seem relatively straightforward, though I did not execute all of them.

**Ethical Considerations:**

No, there are no or only very minor ethics concerns

**Final Justification:**

In my judgement, this is a quality benchmark that holds value in testing the mathematical and analytical capabilities of advanced AI systems such as LLMs. I had some minor concerns about the paper, but those were resolved in the discussion period. I no longer have any concerns about this paper.

I recommend acceptance.

**Limitations Weaknesses:**

I have no major concerns about this paper.

# Minor Concerns

1. The motivation for this benchmark dataset isn't really clear. The first paragraph purportedly gives the motivation for this dataset, but it uses vague buzzwords and doesn't provide a clear explanation for how this sort of benchmark will advance climate science:
> "[LLMs] offers transformative potential for addressing complex challenges in atmospheric science... [Using LLMs for climate-related tasks] requires a robust and comprehensive evaluation framework."

I'm not convinced by these explanations. I see this benchmark more as a way of evaluating LLM progress, not as being "essential" for applying LLMs to atmospheric science problems. We don't necessarily need benchmarks to do that, and performance on benchmarks isn't necessarily meaningful for understanding whether LLMs are useful scientific tools in atmospheric (or any) field of science.

2. Inspired by reference [34], where performance on the GSM8K benchmark drops significantly when the numerical variables are varied, this paper varies numerical values and tests performance. By testing four reasoning models on a benchmark dataset called "ASTMOSSCI-BENCH30", as shown in figure 3 they find that this symbolic perturbation leads to drops in accuracy.

I have serious doubts about the meaningfulness of this evaluation. The whole point of reference [34] was to show that there is data contamination in the GSM8K benchmark. However, this benchmark is not public yet, so (presumably) there couldn't possibly be data contamination. Why test for data contamination if data contamination isn't possible?

Since there almost certainly isn't data contamination in this benchmark, then can the authors please explain why they believe there would be a drop in accuracy? I think the best explanation is the null hypothesis of "random chance". Can the authors show that figure 3 has enough statistical power that they can reject the null hypothesis? If they believe they can reject the null hypothesis, what explains the drop in accuracy?

Furthermore, what do the authors mean by "generalize beyond pattern-matching strategies"? This statement doesn't make sense to me. What pattern matching are the LLMs doing?

Overall, the motivation and purpose of these experiments don't make sense to me, and the conclusions drawn from them don't make sense to me.

3. Reference [15], ClimaX, is not a climate-focused LLM. Please fix this.

# Questions and Comments

1. It is not explained how the 240 questions from MCQ10_EXT are created. Are these 24 questions with varying numerical parameters?
2. Why is MCQ10_EXT separate from MCQ10? Why not put them into a single dataset?
3. The first sentence of the second paragraph ("Constructing a comprehensive benchmark ...") is repetitive. It should probably be cut.
4. The em dashes in the second and third paragaph should probably be periods.
5. Em dashes should not be followed by capitalized letters.
6. Contributions 2 and 3 are not two separate contributions, but are really the same contribution. Analyzing the results of an evaluation is inherently part of a "comprehensive evaluation". Please only list two contributions, not three.
7. On two pages (page 2 and page 19), (vi) is used when the authors really mean (iv).
8. Please explain what "difussion", "confusion", and "randomization" are in figure 1. Difussion is clearly mispelled.
9. Please define OCR.
10. ASTMOSSCI-BENCH30 is never explained. Is it the same as MCQ30?
11. Can you give a little more detail on the MCQ30 dataset? It isn't clear to me what exactly this dataset is and how it was created. How many questions are in the MCQ30 dataset? Are they the same questions as in the MCQ10 dataset, just with different numerical parameters?

**Strengths Contributions:**

* The authors explain how they generated the dataset in detail. Those decisions seem well thought-out and justified.
* The questions are challenging and diverse, covering five subfields related to atmospheric sciences.
* The evaluations are useful for evaluating the performance of LLMs on the benchmarks.
* The results seems pretty straightforward to reproduce using the provided code.

---

> ### Author Rebuttal · Authors · 2025-07-31
>
> We sincerely thank the reviewer for the detailed, thoughtful, and constructive feedback. The comments helped us sharpen the motivation for ATMOSSCI-BENCH, clarify the intent behind symbolic perturbations, and correct several presentation issues.
>
> > **W1: Unclear motivation for this benchmark**
> >
>
> Thanks for raising this excellent point. Our intended message is that a **robust and comprehensive evaluation framework could *advance* the use of LLMs in atmospheric science**, rather than directly applying them at this stage. We will clarify this in the updated version.
>
> That said, we believe benchmarks are crucial for **AI for Atmospheric Science**—and more broadly, for **AI4Science**—for the following reasons:
>
> 1. **Benchmark as a tool to evaluate and improve LLMs**: We agree that this benchmark is more a way of evaluating LLM progress. Benchmarks provide valuable insights into model behavior, help uncover systematic errors, and motivate further research toward improving LLMs for domain-specific reasoning tasks.
> 2. **Contribution to science benchmarks**: Our dataset complements existing science benchmarks by covering a currently underrepresented area—atmospheric science. This helps assess whether LLMs are overly tailored to fields like biology or physics while overlooking others.
> 3. **Benchmark as a proxy for scientific reasoning**: While benchmark performance may not fully reflect whether an LLM is a useful scientific tool, we argue it meaningfully reflects a model’s ability to: a) Understand domain knowledge; b) Perform domain-specific reasoning.
>
> Indeed, ATMOSSCI-BENCH is not designed to test t*h*e utility of LLMs as scientific tools today, but rather to assess whether they are **capable of understanding and solving graduate-level problems** in the domain. These skills are foundational to higher-level tasks such as **proposal writing, data analysis, and experiment automation** — all of which are essential steps toward building an “AI Scientist” [1,2]. Thus, we see this work as a necessary **precursor** to using LLMs as effective tools in atmospheric (or other) sciences.
>
> Our OED-based results also support this perspective, showing that LLMs still have room to grow in reasoning abilities, which this benchmark can help drive forward.
>
> We sincerely appreciate the feedback, which helped us refine and clarify our motivation. We will incorporate the above points into the revision to make our contribution clearer and more compelling.
>
> Reference:
>
> [1] Gottweis, Juraj, et al. "Towards an AI co-scientist." *arXiv preprint arXiv:2502.18864* (2025).
>
> [2] Zheng, Tianshi, et al. "From automation to autonomy: A survey on large language models in scientific discovery." *arXiv preprint arXiv:2505.13259* (2025).
>
> ---
>
> > **W2: Purpose of Symbolic Perturbation & Discussion of Data Contamination**
> >
>
> **Regarding the drop in accuracy under symbolic perturbation:** the question source comes from course materials at the authors’ institute, so we cannot claim the benchmark is entirely free of contamination—a small subset of questions may be slightly derived from some online materials that could already have been exposed to certain LLMs during pre‑training.
>
> The hypothesis of “random chance” is indeed a plausible explanation. To test the null hypothesis that the observed scores will not decrease significantly due to random perturbation of the original questions, if one can reject this null hypothesis, one can confirm that the decrease in score is not a random chance and that data contamination exists in the original questions. We run a hypothesis test and report the resulting z‑scores and p‑values, as shown below:
>
> | Model | z-score | p(z-score) | empirical p |
> | --- | --- | --- | --- |
> | o3-mini | 1.04 | 0.1491 | 0.2667 |
> | Deepseek-R1 | 1.30 | 0.0973 | 0.1667 |
> | QwQ-32B | 2.31 | 0.0105 | 0.0250 |
> | Gemini-2.0-flash-think | 0.63 | 0.2645 | 0.3750 |
>
> Unfortunately, **only** QwQ-32B allows us to reject the null hypothesis (both p < 0.05 significance level). In reference [34], the authors also interpret a deviation of one standard deviation (z-score > 1 but < 2) from the mean accuracy as suggestive of potential data contamination, but they also were not able to fully reject the null hypothesis of “random chance.”
>
> While we cannot conclusively reject the null hypothesis, the fact that the accuracy on the original question set is more than one standard deviation away from the mean is a phenomenon worth attention. It is neither extremely rare (> 2 σ) nor entirely typical (< 1 σ). (Notably, for GPT-o1 on 20 test sets, the accuracy also deviates by one standard deviation.)
>
> **Regarding the purpose of symbolic perturbation:** To evaluate the robustness of reasoning models, we aim to determine whether the models genuinely understand how to solve problems, whether they are capable of true reasoning and generalization, rather than relying on rote memorization or pattern matching.
>
> Reference [34] highlights that instruction models tend to rely heavily on pattern matching. We draw inspiration from that finding and explore whether the same phenomenon applies to reasoning models.
>
> By pattern matching or rote memorization, we mean that LLMs may rely on matching patterns from their training data—finding similar questions or solutions—rather than reasoning through the problem independently.
>
> With the standard deviation away from the mean phenomenon mentioned above, we tend to conclude that the reasoning models evaluated in our benchmark may still be at risk of insufficient robustness under symbolic perturbation, suggesting that they possibly rely on pattern matching instead of genuine reasoning.
>
> **ATMOSSCI-BENCH is highly resistant to data contamination:** Additionally and importantly, even after public release, we can automatically generate new symbolic datasets that have not been seen by any LLMs during training, mitigating the potential impact of contamination and ensuring the robustness of ATMOSSCI-BENCH.
>
> ---
>
> > **W3: ClimaX model description**
> >
>
> We appreciate the reviewer’s attention to this detail. *ClimaX* [15] is not a large‑language model; it is a climate foundation model that learns spatiotemporal representations from multi‑modal reanalysis and remote‑sensing fields. In the updated version, we will replace the phrase “climate‑focused LLM” with “climate foundation model” wherever *ClimaX* is mentioned.
>
> ---
>
> > **Questions and Comments:**
> >
> 1. *Creation of MCQ10_EXT:* MCQ10_EXT is generated in exactly the same way as MCQ10: we apply our MCQ Generation Framework and vary the numerical parameters within physically valid ranges. We will add this clarification at line 167.
> 2. *Separation of MCQ10_EXT separate:* As noted on line 149, our primary focus is Atmospheric Dynamics and Atmospheric Physics. The remaining three domains are auxiliary. Keeping MCQ10_EXT separate lets the overall accuracy reflect performance on the core disciplines without being dominated by the auxiliary ones.
> 3. *Repetitive first sentence of paragraph 2:* Agreed. We will remove the redundant sentence.
> 4. *Replace em‑dashes with periods (paragraphs 2–3):* We will replace those em‑dashes with periods.
> 5. *Capital letters after em‑dashes:* We will lowercase the first letter after each em‑dash.
> 6. *Contributions 2 and 3 appear redundant:* Thanks for the suggestion; we plan to merge them to make the introduction more concise and well-organized.
> 7. *Misnumbered list items ((vi) vs (iv) on pages 2, 19):* We will correct the Roman numerals.
> 8. *Define “difussion”, “confusion”, and “randomization” in Figure 1; fix spelling:* The option‑generation methods are detailed in Appendix F, but we omitted the method names:
>     - **Diffusion** (yes, we misspelled it as “difussion”): randomly swaps two variables in the computation.
>     - **Confusion:** alters one variable to produce a close but incorrect result.
>     - **Randomization:** assigns all variables new values within their constraints, still obeying the rule‑based checks.
>     - **Default (if the other three methods fail)**: A default strategy creates scaled multiples of the correct answer.
>
>     We will correct the spelling and point to Appendix F.
>
> 9. *Define OCR:* OCR (Optical Character Recognition) converts text images into machine‑readable text. We use Mathpix, a state‑of‑the‑art OCR engine, to extract questions and explanations from source materials. We will define that in our updated version.
> 10. *ASTMOSSCI-BENCH30 unexplained:* ASTMOSSCI-BENCH30 is the same as MCQ30; we will correct the typo.
> 11. *More details on MCQ30:* MCQ30 is constructed from the same templates as MCQ10, again utilizing the MCQ Generation Framework, but with different numerical perturbations. We will expand the creation details at line 169 and clarify the total number of questions at line 345.
>
> ---
>
> > **Dataset Code Comments:**
> >
>
> Thank you for highlighting the inconveniences in our code and for taking the time to work through them. We will incorporate your valuable feedback and revise the problematic sections to make the installation process more straightforward and user‑friendly.
>
> ---
>
> We again thank the reviewer for recognizing ATMOSSCI-BENCH as a rigorous, challenging benchmark for testing the symbolic and mathematical abilities of LLMs. This acknowledgment carries particular weight for the atmospheric‑science community: as noted in MC1, the broader scientific AI research is moving toward an *AI co‑scientist* that can assist with hypothesis generation, data analysis, and experimental validation. Enhancing LLMs’ domain knowledge and reasoning is a prerequisite for that vision. We therefore hope ATMOSSCI-BENCH will accelerate progress toward an AI co‑scientist and, ultimately, faster scientific discovery.

---

> > ### Comment · Reviewer_WSaL · 2025-08-02
> > **Reviewer response**
> >
> > Thank you for this thorough and helpful response. I believe my comments and concerns have been addressed appropriately. I do not see the promised changes to the current paper PDF, but I trust that the authors will follow through on their promised changes.
> >
> > **Motivation:** The "intended message" in the rebuttal is much clearer than in the original paper. Please include this discussion in the revised paper.
> >
> > **Symbolic perturbation:** I initially did not understand what the authors meant by "rely on pattern matching instead of genuine reasoning". If I understand correctly now, the authors appear to be using the term "pattern matching" as a synonym for "data contamination". The explanation in the rebuttal is clearer than in the paper, and the explanation for these experiments in the paper should probably be modified to be more explicit about testing for data contamination.
> >
> >  Do you plan to include these p-values in the paper or in an appendix? I believe the paper would benefit from them.
> >
> > **ATMOSSCI-BENCH is highly resistant to data contamination** I wouldn't be so certain about this. There are other ways that datasets can be contaminated, even when parameter values are modified.
> >
> > # Recommendation
> >
> > My concerns were able to be resolved in the discussion period, as expected. I continue to recommend acceptance, though not with high confidence.

---

> > > ### Author Response · Authors · 2025-08-03
> > >
> > > We sincerely thank the reviewer for the constructive follow-up and for recommending acceptance.
> > >
> > > **Promised changes:** We appreciate the reviewer's trust and will implement **all** changes and refinements outlined in our rebuttal in the revised version.
> > >
> > > **Motivation**: Thank you for pointing out that the original manuscript obscured our intended message. We will incorporate the clearer discussion of motivation from the rebuttal into the revised paper so that our contribution is framed unambiguously.
> > >
> > > **Symbolic perturbation:** Yes, we thought that "pattern matching" could be used interchangeably as "data contamination" in our context, but it should be more precise to say "pattern matching" as an indicator for "data contamination". We will revise the manuscript to make this relationship explicit and to explain the perturbation experiments in clearer terms used in the rebuttal. In addition, we will include the p-value analysis (which will be placed in an appendix), which provides quantitative evidence for the effectiveness of the perturbations.
> > >
> > > **Highly resistant to data contamination:** Our symbolic-perturbation study shows that scores drop markedly when superficial cues are altered, demonstrating that ATMOSSCI-BENCH can distinguish models that merely memorise training data from those that genuinely reason. Although we anticipate that new forms of data contamination will emerge, we will continuously update ATMOSSCI-BENCH to address them.
> > >
> > > We are grateful for the reviewer’s endorsement and constructive feedback, and we hope that ATMOSSCI-BENCH will benefit the atmospheric-science, LLM, and broader AI4Science communities.

---

### Official Review · Reviewer_3UYW · 2025-07-02

**Rating:** 4
**Confidence:** 3

**Summary:**

The paper introduces ATMOSSCI-BENCH, a new graduate-level benchmark designed to evaluate the reasoning capabilities of LLMs in the complex domain of atmospheric science. This benchmark features a dual-format design with both multiple-choice (MCQs) and open-ended questions (OEQs) across five domains, using a template-based framework with symbolic perturbation to ensure diversity and test for genuine reasoning over pattern matching.

**Additional Feedback:**

N/A

**Dataset Code Accessibility:**

Yes

**Dataset Code Comments:**

Can be found on kaggle website.

**Ethical Considerations:**

No, there are no or only very minor ethics concerns

**Final Justification:**

The vast majority of the concerns have been addressed after the rebuttal. I have also taken into account the comments from other reviewers and the author's responses, and I believe it is worthwhile to raise the score to a positive one.

**Limitations Weaknesses:**

1. The paper categorizes models into four groups, with a core conclusion being that "reasoning models" significantly outperform others. However, the paper does not provide a deep explanation from a technical or architectural perspective. What is the fundamental difference that separates "reasoning models" (like GPT-o1) from "instruction-tuned models" (like GPT-4o? The current conclusion remains at the level of "what is better" rather than delving into "why it's better."

2. In the evaluation process for OEQs (open-ended questions), using an LLM as an evaluator carries the risk of circular reasoning and bias.

3. The paper observes that domain-specific models like ClimateGPT perform poorly and speculates that the reason is "their pre-training data tends to be non-expert-level text" or they are "designed to act as a research assistant". This is a crude speculation without sufficient supporting evidence. Could it be due to a smaller parameter size? Furthermore, a model like o3-mini achieved an average score of 89%. If the score is this high, does it suggest that the benchmark does not present a significant challenge?

4. By dividing questions into "knowledge-intensive" and "reasoning-intensive," the paper argues that reasoning ability is the bottleneck. A more convincing approach would be to introduce a RAG baseline. If the authors could demonstrate that a basic model equipped with a RAG system (capable of perfectly retrieving all relevant knowledge) still performs worse than a top-tier reasoning model without RAG, the conclusion that "reasoning is more important than knowledge" would be rock-solid. The current paper lacks this component, which slightly diminishes the persuasiveness of this important conclusion. Additionally, would newer RAG frameworks like DualRAG, which integrate a reasoning workflow, perform better? Such approaches seem to combine the advantages of being both knowledge-intensive and having strong reasoning capabilities.

5. Detail issue: On page 2, in the "Contribution 2" section, the fourth category of models should be labeled (iv), not (vi).

**Strengths Contributions:**

It is crucial to evaluate the capabilities of LLMs in key scientific fields such as climate change. This paper accurately addresses a gap in this research area and holds significant practical and academic value.

The use of graduate-level materials ensures the challenging nature of the problems. The method of symbolic perturbation draws from previous work and is successfully applied to a new domain, serving as an effective means to test a model's true generalization and reasoning abilities rather than rote memorization. The combination of MCQs and OEQs also allows for a more comprehensive evaluation.

The evaluation of relevant existing models is thorough.

---

> ### Author Rebuttal · Authors · 2025-07-31
>
> We sincerely thank the reviewer for the thorough and insightful evaluation of our work.
>
> ---
>
> > **W1: Deep explanation from a technical or architectural perspective**
> >
>
> We agree that adding this perspective would elevate the depth and clarity of our conclusions. Here is a deeper explanation from a technical and architectural perspective regarding the **fundamental differences** that separate "reasoning models" (e.g., GPT-o1) from "instruction-tuned models" (e.g., GPT-4o):
>
> 1. **Deliberate “Thinking” Stage + Extra Inference Compute:** Reasoning models insert a dedicated *“<think>”* phase and allocate additional compute at inference time, enabling long chain‑of‑thought processing that lets smaller reasoning models outperform much larger instruction‑tuned ones on complex tasks [1, 2, 3].
> 2. **Self‑Checking to Limit Error Propagation:** Through reinforcement learning, models develop emergent behaviors—self‑verification, reflection, and back‑tracking—that catch and correct faulty reasoning steps before the final answer [1, 3].
> 3. **Specialized RL Post‑Training for Reasoning:** Models like DeepSeek‑R1 rely on pure RL fine‑tuning with **Group Relative Policy Optimization (GRPO)** on step‑by‑step solution data, explicitly rewarding correct intermediate reasoning and strengthening overall chain‑of‑thought quality [3, 4].
>
> For the reasons above, reasoning models show significant improvements in complex **multi-step** reasoning tasks. As shown on line 88, all reasoning models achieve substantial gains on benchmarks tailored for such tasks [1, 2, 4].
>
> ATMOSSCI-BENCH is specifically designed to evaluate whether LLMs equipped with thinking stages, intermediate learning, and self-correction can effectively solve multi-step problems. The results confirm that ATMOSSCI-BENCH successfully distinguishes reasoning models from instruction-tuned models in Atmospheric Science, thus demonstrating its ability to assess reasoning capacity while posing meaningful challenges for future LLM development.
>
> Reference:
>
> [1] OpenAI. *Learning to Reason with LLMs*. OpenAI, 2024. Available at: https://openai.com/index/learning-to-reason-with-llms/
>
> [2] Google DeepMind. *Gemini Thinking*. Google, 2024. Available at: https://ai.google.dev/gemini-api/docs/thinking
>
> [3] Guo, Daya, et al. "Deepseek-R1: Incentivizing Reasoning Capability in LLMs via Reinforcement Learning." *arXiv preprint* arXiv:2501.12948, 2025. Available at: https://arxiv.org/abs/2501.12948
>
> [4] Qwen Team. *QwQ-32B: Embracing the Power of Reinforcement Learning*. QwenLM, 2024. Available at: https://qwenlm.github.io/blog/qwq-32b/
>
> ---
>
> > **W2: LLM-as-Judge bias**
> >
>
> LLM‑based evaluation is increasingly accepted in scientific benchmarks, and recent studies show strong agreement with expert human assessments, while observing that any remaining bias has now diminished to an acceptable level. To bolster confidence in our evaluation pipeline, we conducted a human-LLM agreement study on open-ended questions. These results indicate that **LLMEvaluator** is sufficiently reliable for our study. Space constraints prevent a full discussion here; please refer to our response to Reviewer qRMF (Section W1: Clarification on LLM‑as‑Judge Evaluation for OEQ) for complete details.
>
> ---
>
> > **W3: Concern about speculation on the causes of domain‑specific model under‑performance**
> >
>
> Thanks for highlighting the need for clearer evidence regarding why domain‑specific models underperform. We first address the possibility that model size is the root cause. On the MCQ‑10 benchmark, the base model **Qwen 2.5‑72B‑Instruct‑Turbo** achieves an accuracy of **57.01 %**, while the identically‑sized **GeoGPT‑Qwen 2.5‑72B** scores **34.93 %**. Because both models share the same 72 billion parameter count yet differ by more than twenty percentage points, model scale alone cannot explain the performance gap.
>
> To further explore the underperformance of domain‑specific models, we conducted an error analysis for GeoGPT‑Qwen 2.5‑72B on the MCQ‑10 dataset. We manually identified the errors for each question. The error types are the same as Appendix I, with one additional category, V. Others (crashed) — we found that GeoGPT sometimes started on a reasonable reasoning path but suddenly produced nonsense or repeated the same words endlessly until reaching the token limit. The results are summarized in the table below (Note that one question may exhibit multiple error types, so the aggregated percentage is not 100%.):
>
> | I. Lack of Relevant Knowledge | II. Incorrect Calculation | III. Misunderstanding the Question | IV. Faulty Reasoning | V. Others (crashed) |
> | --- | --- | --- | --- | --- |
> | 57% | 38% | 30% | 62% | 44% |
>
> These error patterns are consistent with our explanation of GeoGPT’s. Indeed, GeoGPT’s stated design goal: it is intended “to understand the demands of non‑professional users, then think, plan, and execute defined GIS tools to produce effective results.” Optimizing for tool planning rather than sustained step‑by‑step problem‑solving data, therefore, provides a more plausible explanation for its lower accuracy than parameter size.
>
> Regarding ClimateGPT, we agree that our earlier wording was imprecise. ClimateGPT’s instruction‑tuning corpus is drawn almost entirely from crowd‑sourced prompt‑completion pairs and multi‑turn dialogues. We should state, “their pre-training data are not for structured, step‑wise reasoning found in scholarly problem solving” instead.
>
> Taken together, the evidence suggests that the principal limitations of current domain-specific LLMs lie not in their parameter count but in their training objectives and data composition: a focus on tool interaction or conversational style can leave gaps in both domain knowledge breadth and reasoning robustness.
>
> ---
>
> > **W4:  Dual‑Format Design Keeps the Benchmark Challenging**
> >
>
> Indeed, o3‑mini achieves an average accuracy of 89 % on the MCQ portion—nearly saturated—yet scores below 40 % on the OEQ test set (Table 4), confirming that ATMOSSCI-BENCH remains challenging. This result reflects the purpose of our dual‑format design. MCQs enable automated scoring, support controlled symbolic perturbations, and ensure unambiguous evaluation; they also play a crucial role in distinguishing reasoning models from non-reasoning baselines and in structuring our experiments. However, as noted on line 125, prior studies have shown that MCQs may let models earn high marks through answer recognition or pattern matching rather than genuine conceptual understanding [29, 31, 37]. By contrast, OEQs—whose output space is unconstrained and lacks a single universal metric—better reveal true reasoning capabilities, thereby elevating the benchmark’s overall difficulty.
>
> ---
>
> > **W5: Reasoning vs. Knowledge—RAG Baseline Needed**
> >
>
> We appreciate the reviewer’s insight that simply splitting questions into “knowledge‑intensive” and “reasoning‑intensive” categories is not sufficiently rigorous. Following the suggestion to add a retrieval‑augmented generation (RAG) baseline, we indexed all available course materials into a vector database and quickly discovered that naïve retrieval failed to surface much relevant knowledge. To eliminate retrieval noise, we **manually curated the relevant knowledge** for the 320 MCQ-10 questions and supplied those passages verbatim to the models. This isolates the effect of *reasoning* from *knowledge access*:
>
> | Model | Relevant knowledge provided | Accuracy (%) |
> | --- | --- | --- |
> | GPT-o3-mini | true | 77.5 |
> | GPT-o3-mini | false | 73.8 |
> | GPT-4o | true | 53.75 |
> | GPT-4o | false | 52.81 |
>
> Both models improve with perfect knowledge, confirming that knowledge gaps exist. Crucially, **GPT‑o3‑mini without any external knowledge still outperforms GPT‑4o even when GPT‑4o is fed all the relevant passages by nearly 20 %.** Moreover, adding knowledge increases GPT-o3-mini by ~4%, but GPT‑4o by only ~1%, indicating that stronger reasoning enables a model to exploit knowledge far more effectively. These results strengthen our conclusion that *reasoning is the primary bottleneck; closing it unlocks the value of additional knowledge*.
>
> **Challenge of RAG in atmospheric science?**
>
> 1. **Fragmented, context‑dependent literature.** A single paper often presents several closely related formula variants; embeddings blur these distinctions, and many excerpts omit prerequisite definitions, hampering standalone retrieval.
> 2. **Complex relational structure.** Unlike simple fact triples (e.g., “James Cruze → birth year → 1884”), atmospheric knowledge involves intertwined equations, approximations, and causal chains. Emerging graph-based RAGs (e.g., GraphRAG) could help with capturing complex structure, but constructing a high-quality atmospheric-science graph demands extensive expert curation.
>
> **Discussion about DualRAG**
>
> DualRAG’s iterative cycle of reasoning-augmented querying and progressive knowledge aggregation is a compelling way to scale performance, allowing strong reasoning and high-quality knowledge to reinforce each other. Currently, the primary obstacle is the lack of a good domain-specific knowledge database or graph for atmospheric science; building this resource is a crucial first step before advanced RAG frameworks can be rigorously evaluated. Once such a graph exists, we are confident that pairing it with DualRAG will unlock still greater gains in LLM capability.
>
> We will incorporate the RAG experiment and a discussion of DualRAG into the updated version.
>
> ---
>
> For the details, thanks for pointing out this typographical error. We will correct the label from “(vi)” to “(iv)” in the “Contribution 2” section on Page 2 in the updated version.
>
> ---
>
> We appreciate the reviewer for valuable suggestions to enhance ATMOSSCI-BENCH’s evaluative rigor and persuasiveness. We are grateful for the reviewer’s guidance, which has markedly improved the quality and depth of our work, and we welcome any further suggestions.

---

> > ### Comment · Reviewer_3UYW · 2025-08-04
> >
> > Thank you to the authors for the detailed response. The new data and analysis have resolved the majority of my concerns, and therefore I will raise my score to a positive one.

---

> > > ### Author Response · Authors · 2025-08-05
> > >
> > > We appreciate your careful reconsideration. We’re pleased that the new data and analyses resolved most concerns and will reflect these updates in the updated version.

---

### Official Review · Reviewer_gtVK · 2025-07-03

**Rating:** 5
**Confidence:** 4

**Summary:**

This paper introduces AtmosSci-Bench, a benchmark designed to systematically evaluate the performance of Large Language Models (LLMs) in atmospheric science. The benchmark features a dual-format design, incorporating both Multiple-Choice Questions (MCQs) and Open-Ended Questions (OEQs), spanning five core categories: hydrology, atmospheric dynamics, atmospheric physics, geophysics, and physical oceanography. The MCQ questions are generated by a template-based framework, which allows for symbolic perturbation to create diverse, graduate-level problems, mitigating issues of pattern matching from training data. The authors conduct an extensive evaluation of various LLMs, categorizing them into instruction-tuned, advanced reasoning, math-augmented, and domain-specific climate models. The findings highlight the superior performance of reasoning models and underscore the benchmark's effectiveness in differentiating LLM capabilities, particularly in complex scientific problem-solving.

**Dataset Code Accessibility:**

Yes

**Dataset Code Comments:**

Code and dataset are provided

**Ethical Considerations:**

No, there are no or only very minor ethics concerns

**Final Justification:**

I appreciate the authors for thoroughly addressing each of my concerns. I am also satisfied by the rebuttals to the other reviews. I will raise my score from 4 (borderline accept) to 5 (accept).

**Limitations Weaknesses:**

I identified 3 main limitations and weaknesses, and I would be happy to raise my score if these are addressed.

**LW1) Lacking discussion about cost of dataset creation and model evaluation**

Especially since the paper evaluates API-gated models, I believe the paper would be made stronger by disclosing more information about evaluation costs incurred for the OpenAI, Gemini, and DeepSeek models. This information helps improve reproducibility as well.

**LW2) Limited evaluation of domain-specific models**

The paper notes that domain-specific models like ClimateGPT and GeoGPT underperform the other models tested, and the authors attribute this to potential biases in pretraining data or design for tool utilization rather than direct reasoning. To better understand the flaws of these domain-specific models, I would find it useful to see an error analysis (similar to Appendix I) conducted for these domain-specific models.

**LW3) Limited evaluation of LLM-as-Judge validity**

The authors note that QuantityEvaluator and ExpressionEvaluator, which are more accurate and grounded, are in over 90% of questions. The remaining <10% is evaluated by LLMEvaluator, which is based on GPT-4o-mini. While the description of LLMEvaluator in the paper seems reasonable, I would like to see a comparison of LLMEvaluator vs. QuantityEvaluator and ExpressionEvaluator on the questions that QuantityEvaluator and ExpressionEvaluator were able to adjudicate. If LLMEvaluator agrees with QuantityEvaluator and ExpressionEvaluator on most instances, then this inspires more confidence in the accuracy of LLMEvaluator.

**Strengths Contributions:**

**SC1) Novelty of the Benchmark**

The creation of AtmosSci-Bench fills a gap in LLM evaluation by providing a specialized and rigorous benchmark for atmospheric science. As highlighted in the introduction, atmospheric science problems are inherently interdisciplinary and require integration of theoretical modeling with real-world phenomena, making existing general benchmarks insufficient. AtmosSci-Bench's tailored design directly addresses this complexity. The novelty is further reflected by the poor performance of LLMs instruction fine-tuned for math problems, compared to reasoning-based LLMs.

**SC2) Dual-Format Question Design**

The dual-format approach, combining MCQs and OEQs, is a strength. MCQs enable scalable, automated assessment with symbolic templates, while OEQs probe deeper conceptual understanding and reasoning ability in less constrained settings.

**SC3) Robust MCQ Generation Framework with Symbolic Perturbation**

The template-based MCQ generation framework, inspired by GSM-Symbolic, replaces fixed numerical values with variable placeholders and applying rule-based constraints. The benchmark helps avoid pattern matching from potentially contaminated training data. This enhances the benchmark's discriminative power and robustness.

**SC4) Comprehensive LLM Evaluation and Insightful Findings**

The paper conducts a thorough evaluation across a wide range of representative LLMs. The analysis yields interesting and important insights, such as the superior performance of reasoning models over instruction, math, and even domain-specific models. The observation regarding inference time scaling and diminishing returns and the sensitivity to symbolic perturbations are particularly valuable for guiding future LLM development.

**SC5) Well-Written and Organized Presentation**

The paper is well-written, clearly organized, and easy to understand.

---

> ### Author Rebuttal · Authors · 2025-07-31
>
> We thank the reviewer for the thoughtful and constructive feedback. Below, we address each of the three identified weaknesses (W1–W3) and describe the concrete revisions incorporated into the updated version.
>
> > **W1:  Lacking discussion about cost of dataset creation and model evaluation:**
> >
>
> We fully agree that reporting the *exact* cost of dataset construction and model evaluation strengthens reproducibility. We have therefore added a dedicated “Cost & Runtime” section with the updated version. Following are the details of this section:
>
> **Cost of dataset construction**: In our automatic problem solver, we utilize the GPT‑4o web interface under a fixed‑price subscription. Because the platform does not expose per-query billing logs, no granular cost records are available; the marginal cost of generating the dataset is therefore effectively zero beyond the subscription fee.
>
> **Cost of model evaluation**: We now disclose the exact token counts, parallel‑invocation settings, wall‑clock runtimes, and USD costs for every API‑based LLM (Table R1) and the runtimes for all GPU‑hosted models (Table R2).
>
> **Table R1  API‑based evaluation cost on MCQ10**
>
> | Model | Prompt Tokens (M) | Reasoning Tokens (M) | Output Tokens (M)† | Cost (USD) | Runtime (min) | Parallel size |
> | --- | --- | --- | --- | --- | --- | --- |
> | deepseek-R1 | 0.2 | 4.3 | 4.6 | 40 | 216/24,526†† | 32/1††  |
> | deepseek-V3 | 0.2 | - | 0.6 | 10 | 213/232 | 16/2 |
> | GPT-o1 | 0.2 | 2.4 | 2.5 | 200 | 231 | 8 |
> | GPT-4o | 0.2 | - | 0.3 | 5 | 33 | 8 |
> | GPT-o3-mini | 0.2 | 2.2 | 2.8 | 15 | 137 | 4 |
> | GPT-4o-mini | 0.2 | - | 0.6 | 1 | 30 | 10 |
> | gemini-2.0-flash-thinking-exp-01-21 | 0.2 | untrackable | 0.5 (Reasoning Token Exclude) | free but have daily limit | 204 | 5 |
> | QwQ-32B-Preview_32K (TogetherAPI) | 0.2 | untrackable | 3.6 | 10 | 276 | 10 |
> | Qwen2.5-72B-Instruct-Turbo (TogetherAPI) | 0.2 | - | 0.9 | 2 | 44 | 30 |
> | Llama-3.1-405B-Instruct-Turbo(TogetherAPI) | 0.2 | - | 0.5 | 3 | 163 | 64 |
>
> † Unless noted, output tokens include reasoning tokens.
>
> †† “216 / 24,526” means 32‑way parallel execution finishes in 216 min; single‑thread execution would take 24,526 min.
>
> **Table R2  GPU‑hosted evaluation runtime:**
>
> |  | Runtime (min) | GPUs |
> | --- | --- | --- |
> | Qwen2.5-Math-1.5B-Instruct | 133 | 8*RTX4090 |
> | Qwen2.5-Math-7B-Instruct | 467 | 8*RTX4090 |
> | Qwen2.5-Math-72B-Instruct | 3051 | 8*RTX4090 |
> | Qwen2.5-3B-Instruct | 123 | 8*RTX4090 |
> | Qwen2.5-7B-Instruct | 141 | 8*RTX4090 |
> | climategpt-7b | 783 | 8*RTX4090 |
> | climategpt-70b | 2354 | 8*RTX4090 |
> | gemma-2-27b-itLlama-3.1-405B-Instruct-Turbo | 250 | 8*RTX4090 |
> | Qwen2.5-72B-GeoGPT  | 2,880 | 4*A800 |
>
> These tables presents the statistics of various LLMs on the MCQ10 dataset under standardized experimental settings. Please use this as a reference when estimating usage for other datasets, with the following multipliers: MCQ30: ×3, MCQ10_EXT: ×0.35, OEQ: ×0.6.
>
> Please note that only key models are displayed. For models not listed, one can refer to those with the same API provider and similar configuration parameters.
>
> We hope this additional detail resolves the reviewer’s concern and enhances the paper’s reproducibility.
>
> ---
>
> > **W2: Limited evaluation of domain-specific models:**
> >
>
> As suggested, an error analysis of domain‑specific models is very useful. We conducted an error analysis for GeoGPT‑Qwen 2.5‑72B on the MCQ‑10 dataset. We manually identified the errors for each question. The error types are the same as Appendix I, with one additional category, V. Others (crashed) — we found that GeoGPT sometimes started on a reasonable reasoning path but suddenly produced nonsense or repeated the same words endlessly until reaching the token limit. The results are summarized in the table below (Note that one question may exhibit multiple error types, so the aggregated percentage is not 100%.):
>
> | I. Lack of Relevant Knowledge | II. Incorrect Calculation | III. Misunderstanding the Question | IV. Faulty Reasoning | V. Others (crashed) |
> | --- | --- | --- | --- | --- |
> | 57% | 38% | 30% | 62% | 44% |
>
> The high incidence of *Faulty Reasoning* (62 %) suggests that GeoGPT‑Qwen 2.5‑72B often assembles superficially plausible chains of thought yet fails to maintain logical consistency to the end of a problem. It indicates the model was trained more on specialized terms than on step‑by‑step reasoning. Likewise, the substantial share of *Lack of Relevant Knowledge* (57 %) indicates that the model’s pre‑training corpus—although focused on geoscience—does not sufficiently cover the breadth of background facts required by the MCQ‑10 benchmark, which mixes atmospheric physics, climatology, and numerical methods. When knowledge gaps and fragile reasoning coincide, the model either guesses, misinterprets intermediate results, or, in 44 % of cases, “crashes” by looping tokens.
>
> Together, these patterns support our hypothesis in the main paper: current domain‑specific LLMs risk overfitting to narrow stylistic cues in their training data and lack the robust reasoning scaffolding seen in general‑purpose “reasoning models.” To close this gap, future work could (i) augment geoscience corpora—adding atmospheric‑science knowledge, numerical‑methods theory, and worked problem‑solving approaches, all with curated step‑by‑step solutions; and (ii) incorporate reasoning‑oriented fine‑tuning objectives (e.g., chain‑of‑thought supervised signals).
>
> ---
>
> > **W3: Limited evaluation of LLM-as-Judge validity:**
> >
>
> We wish to clarify that, while **QuantityEvaluator** and **ExpressionEvaluator** are highly accurate and well‑grounded for *positive* true answers, they may still produce false negatives due to the evaluation in Python library **SymPy**. Mitigating false negatives is one of the design intentions of our **cascade of evaluators,** which **LLMEvaluator** plays as a solid **backup Evaluator** handling potential positive cases that **ExpressionEvaluator** marks as false, thereby reinforcing overall robustness.
>
> Because the accuracy of **LLMEvaluator** is critical, we further validated it through a human grader agreement test. We conducted a human‑LLM agreement study on OEQ problems evaluated by **LLMEvaluator**. Agreement is the percentage of decisions where the LLM’s `is_correct` label matched that of a human grader. Here is the statistic:
>
> | Answer Model | GPT-4o-mini vs Human |
> | --- | --- |
> | deepseek-r1 | 92.79% |
> | o3-mini | 93.02% |
>
> The result shows that > 92 % for GPT‑4o‑mini, which we therefore consider LLMEvaluator sufficiently reliable for our evaluation.
>
> **False negatives example:** Additionally, here is the false negatives example that may occur in ExpressionEvaluator when the LLM output includes extra symbols or text that confuse the Python library **SymPy**. For example:
>
> ```
> "expected_answer": "$\\bar{M}=28.71 \\mathrm{~g} \\mathrm{~mol}^{-1}$",
> "llm_answer": "\\bar{M} \\approx 28.72\\ \\text{g/mol} \\quad\\text{or}\\quad 0.02872\\ \\text{kg/mol}.",
> ```
>
> Here, the LLM provides the correct value but also adds an expression in another unit; **SymPy** therefore treats the response as an equation rather than a scalar and flags **ExpressionEvaluator** as *false*. Importantly, **when SymPy returns a positive match, its verdict is virtually error‑free**.
>
> ---
>
> In summary, the newly added **Cost & Runtime** section, the detailed error analysis of GeoGPT‑Qwen 2.5‑72B, and the expanded human‑LLM agreement study for **LLMEvaluator** directly resolve W1–W3. These revisions, together with the benchmark’s original contributions—a dual‑format dataset, a symbolic‑perturbation MCQ framework, and the first systematic comparison of reasoning‑oriented, math‑augmented, and domain‑specific LLMs in atmospheric science—substantially strengthen the paper’s technical rigor and reproducibility. We appreciate the reviewer’s insights and believe the manuscript is now markedly improved.

---

> > ### Comment · Reviewer_gtVK · 2025-08-03
> > **Thorough rebuttal**
> >
> > I appreciate the authors for thoroughly addressing each of my concerns. I am also satisfied by the rebuttals to the other reviews. I will raise my score from 4 (borderline accept) to 5 (accept).

---

> > > ### Author Response · Authors · 2025-08-05
> > >
> > > Thank you for the thoughtful review and the positive reassessment. We’re glad our responses addressed your concerns, and we will incorporate the clarifications into the revised manuscript.

---

### Official Review · Reviewer_qRMF · 2025-07-04

**Rating:** 5
**Confidence:** 3

**Summary:**

This paper introduces a novel benchmark designed to evaluate the performance of LLMs in atmospheric science. The benchmark covers five core domains in atmospheric science, including hydrology, atmospheric dynamics, atmospheric physics, geophysics, and physical oceanography, and questions are in the formats of multiple choices (MCQs) and open-ended questions (OEQs). The paper conducts a comprehensive evaluation of LLMs. The findings indicate that reasoning models are better on these tasks, but also highlight performance gaps and robustness issues for all LLMs.

**Dataset Code Accessibility:**

Yes

**Dataset Code Comments:**

Code and data URLs are included in submission.

**Ethical Considerations:**

No, there are no or only very minor ethics concerns

**Final Justification:**

I appreciate the authors’ responses, which have addressed my concerns. In my view, this work presents a meaningful contribution, offering a valuable benchmark for evaluating the mathematical and analytical abilities of AI systems.
I maintain my positive score and recommendation for acceptance.

**Limitations Weaknesses:**

While the paper overall seems to have good quality, I have some minor concerns.

- OEQ evaluation with the LLM-as-judge can potentially be biased and inaccurate. Even though previous works have used similar evaluation method and SOTA models like GPT-4o-mini can be strong evaluators, the results would benefit from further validation with human graders to assess whether the automatic evaluators match expert understanding, especially for borderline or multi-step answers.

- No discussion on data contamination. The data is collected from course materials used in classes at the author's university. Could this mean this data has been included in training of some LLMs? I think the symbolic perturbations help mitigate this, but still there is no explicit discussion of benchmark contamination.

**Strengths Contributions:**

- the proposed benchmark targets the complex and interdisciplinary domain of atmospheric science, unlike prior benchmarks, e.g., SciBench and PHYSICS, which focus on general science or physics but lack coverage of atmospheric science.

- The coverage of language models is relatively thorough (> 25 LLMs). There are also two types of questions. The symbolic MCQ generation ensures question diversity and robustness, and the OEQs investiage deeper abilities of LLMs.

- Analysis reveals that domain-specific models underperform reasoning-aligned models, which is an interesting finding that suggests reasoning ability is more critical than domain fine-tuning for solving real-world atmospheric science problems.

- Code and data are opensourced for reproducibility.

---

> ### Author Rebuttal · Authors · 2025-07-31
>
> We sincerely thank the reviewer for the thoughtful, constructive, and encouraging assessment of our manuscript. The goal of the reviewers. At the same time, we provide feedback to address the specific concerns regarding OEQ evaluation bias and potential data contamination:
>
> > **W1: Clarification on LLM-as-Judge Evaluation for OEQ**
> >
>
> Thanks for raising this important point. We agree that the use of LLM-as-judge for open-ended question (OEQ) evaluation can introduce potential biases, especially in cases involving borderline or multi-step reasoning. While prior work has employed similar automatic evaluation strategies (as we mentioned on line 241), we recognize the value of validating such methods against expert human judgments.
>
> To address this concern, we conducted a human-LLM evaluation consistency study over OEQ problems that were handled by LLMEvaluator. Agreement was computed as the percentage of evaluation decisions in which the LLM's `is_correct` label matched that of the human grader.
>
> | Answer Model | GPT-4o-mini matches Human |
> | --- | --- |
> | deepseek-r1 | 92.79% |
> | o3-mini | 93.02% |
>
> These results suggest that LLM-as-Judge achieves >92% agreement with human graders, which we believe is sufficient to serve as a good automatic evaluator in lieu of human annotation.
>
> We appreciate the constructive feedback and agree that human validation is a critical step toward improving the reliability of LLM-based evaluation.
>
> ---
>
> > **W2: Data contamination**
> >
>
> Discussion of data contamination is indeed critical, so we have added a dedicated paragraph.
>
> The question source comes from course materials at the authors’ institute, so we cannot claim the benchmark is entirely free of contamination—a small subset of questions may be slightly derived from some online materials that could already have been exposed to certain LLMs during pre‑training.
> Furthermore, to quantitatively answer this problem, we conduct an additional symbolic‑perturbation experiment based on Figure 3, to test whether the decreasing score of perturbation question sets is caused by “random chance”. Therefore, we test the null hypothesis that the observed scores will not decrease significantly by random perturbation of the original questions  — if one can reject this null hypothesis, one can confirm the existence of data contamination in the original questions. We run a hypothesis test and report the resulting z‑scores and p‑values:
>
> | Model | z-score | p(z-score) | empirical p |
> | --- | --- | --- | --- |
> | o3-mini | 1.04 | 0.1491 | 0.2667 |
> | Deepseek-R1 | 1.30 | 0.0973 | 0.1667 |
> | QwQ-32B | 2.31 | 0.0105 | 0.0250 |
> | Gemini-2.0-flash-think | 0.63 | 0.2645 | 0.3750 |
>
> Noted that **only** QwQ‑32B crosses the conventional 95 % threshold (both p < 0.05 significance level or z-score > 2), allowing us to reject the null hypothesis. Reference [34] (GSM-Symbolic: Understanding the Limitations of Mathematical Reasoning in Large Language Models) likewise treats a deviation of one standard deviation (1 < z-score < 2) from the mean accuracy as suggestive of contamination, yet cannot fully rule out chance either.
>
> Accordingly, while we cannot conclusively reject the null hypothesis, the fact that several models’ accuracies lie more than one standard deviation above the mean remains noteworthy: the phenomenon is neither extremely rare (> 2 σ) nor completely routine (< 1 σ). (For GPT‑o1 over 20 splits, the deviation is also ≈ 1 σ.)
>
> We want to highlight that our benchmark is highly resistant to contamination based on symbolic extensions: even after public release, we can automatically regenerate fresh, symbolically equivalent datasets that no LLM has encountered, thus preserving the benchmark’s robustness.
>
> ---
>
> In summary, we have (1) validated the reliability of LLM‑as‑Judge scoring through a new human‑agreement study, and (2) added a data‑contamination analysis with hypothesis tests in the updated version. These additions strengthen the empirical rigor of *ATMOSSCI-BENCH* while preserving its core contribution: a scalable, domain‑rich testbed that distinguishes the reasoning capacity of LLMs in real‑world atmospheric problems. We are grateful for the insightful feedback, further reinforcing the benchmark’s value to both the LLM and atmospheric science communities.

---

### Note · Authors · 2025-08-12

We sincerely thank all reviewers for their active engagement, thoughtful feedback, and constructive suggestions, which have been invaluable in improving the clarity, rigor, and overall contribution of our work, and for the time and effort that directly helped make the paper more solid and impactful.

The reviewers have agreed on the following key contributions of our benchmark:

1. Introduced the first reasoning-focused benchmark for atmospheric science, evaluating over 25 LLMs in 4 categories.
2. Filled a gap in science benchmarks by covering the underrepresented domain of atmospheric science.
3. Designed dual-format questions (MCQ + OEQ) with a cascade evaluation pipeline, using LLM-as-judge as a reliable backup, to balance scalability, deep reasoning assessment, and robust scoring.
4. Developed a symbolic-perturbation MCQ framework to generate contamination-resistant datasets.

During the rebuttal, we also carefully addressed every raised concern and made substantial improvements to the benchmark and manuscript. Specifically, we:

1. **Added Human–LLM Agreement Study** – confirmed >92% agreement, validating OEQ scoring reliability.
2. **Tested for Data Contamination** – found minimal evidence; benchmark remains contamination-resistant.
3. **Explained Benchmark Design and Reproducibility Enhancements** – clarified reasoning-vs-instruction model differences, symbolic perturbation purpose, benchmark motivation, and added cost/runtime details for transparency.
4. **Analyzed Model Performance with RAG Baseline** – identified faulty reasoning and knowledge gaps in domain-specific models, and used a curated-knowledge RAG baseline to show reasoning strength is more critical than knowledge access.
5. **Fixed Typos and Clarity Issues** – corrected errors, improved definitions, and refined dataset/code descriptions.

We believe that the revised manuscript, together with the strengthened benchmark, addresses all reviewer concerns and provides a more robust and valuable contribution to the community. We will implement all changes and refinements outlined in our rebuttal in the revised version. Once again, we deeply appreciate the reviewers’ and AC’s great help in improving the quality of our paper.

---

### Decision · Program_Chairs · 2025-09-18

**Decision:**

Accept (poster)

**Comment:**

The paper introduces a benchmark designed to evaluate large language models (LLMs) in the field of atmospheric science, covering five key domains. It presents a thorough evaluation of LLM performance, with findings suggesting that reasoning-focused models perform better on these tasks. However, the paper also highlights significant performance gaps and robustness concerns across all LLMs. While there were initial concerns regarding the depth of the explanations and additional results, these issues were effectively addressed during the rebuttal period. After reviewing the paper, the reviewers' comments, and the rebuttal, I am in agreement with the general consensus and recommend accepting this paper.